# Uniting functional network topology and oscillations in the fronto-parietal single unit network of behaving primates

Benjamin Dann[1], Jonathan A Michaels[1], Stefan Schaffelhofer[1], Hansjörg Scherberger[1,2]*

[1]Neurobiology Lab, German Primate Center, Göttingen, Germany; [2]Faculty of Biology, Georg-August University Göttingen, Göttingen, Germany

**Abstract** The functional communication of neurons in cortical networks underlies higher cognitive processes. Yet, little is known about the organization of the single neuron network or its relationship to the synchronization processes that are essential for its formation. Here, we show that the functional single neuron network of three fronto-parietal areas during active behavior of macaque monkeys is highly complex. The network was closely connected (small-world) and consisted of functional modules spanning these areas. Surprisingly, the importance of different neurons to the network was highly heterogeneous with a small number of neurons contributing strongly to the network function (hubs), which were in turn strongly inter-connected (rich-club). Examination of the network synchronization revealed that the identified rich-club consisted of neurons that were synchronized in the beta or low frequency range, whereas other neurons were mostly non-oscillatory synchronized. Therefore, oscillatory synchrony may be a central communication mechanism for highly organized functional spiking networks.

*For correspondence: hscherb@gwdg.de

**Competing interests:** The authors declare that no competing interests exist.

## Introduction

Perception, cognition, and movement are generated by the functional interaction of neuronal circuits. In order to understand the basis of these processes, especially in highly complex networks such as the primate brain, it is essential to know their network structure, termed topology. Graph theoretical approaches have enabled analysis of the brain's network topology (*Watts and Strogatz, 1998*; *Bullmore and Sporns, 2009*). Using such approaches in EEG, MEG, DTI or fMRI studies, anatomical regions have been grouped into functional and anatomically strongly connected modules, which are segregated from each other (*Bullmore and Sporns, 2009*). Still, every region can be reached by bypassing a few others (small-world), a topology which is robust and allows efficient information processing (*Hilgetag et al., 2000*; *Stephan et al., 2000*; *Bullmore and Sporns, 2009*). A few regions of the brain are highly connected and centrally located within the network (*van den Heuvel and Sporns, 2013a*) (hubs) as well as strongly connected to each other (*van den Heuvel et al., 2012*) (rich-club). This rich-club forms a global communication pathway across the network, thereby cross-linking segregated modules (*van den Heuvel and Sporns, 2013b*).

However, single neurons and their functional network topology are the fundamental computational structure of the primate brain. While neuronal modules, hubs, and rich-club organization have been shown in organotypic slices of rats (*Bonifazi et al., 2009*; *Shimono and Beggs, 2014*; *Schroeter et al., 2015*), hardly anything is known about single neuron network topology in the intact brain during behavior. Limitations in recording high number of single neurons in parallel, incorporating distance-dependent connectivity, and addressing subsampling and firing rate biases makes it

**eLife digest** The network of neurons in our brain generates all of our actions, yet it is not well understood how these neurons coordinate their activity with each other. Rhythmic electrical activity that happens at the same time across many different neurons is thought to be crucial for allowing different areas of the brain to communicate. However, it is still unclear what purpose rhythmic activity serves for communication. Are there groups of 'hub' neurons in different brain regions that coordinate overall activity by rhythmically synchronizing the network of neurons? Or is rhythmic activity insignificant for network coordination?

Dann et al. trained three monkeys to follow specific instructions to grasp a handle in different ways. While the monkeys performed the task, the activity of about 100 neurons was recorded simultaneously in three brain regions that are involved in planning and carrying out grasping movements. This revealed that the activity of the neurons was coordinated by a group of strongly connected hub neurons, which were distributed across all three of the brain regions. Nearly all of the hub neurons were rhythmically synchronized with each other, and also communicated with other neurons using rhythmic electrical activity.

Overall, the results presented by Dann et al. suggest that rhythmically synchronized activity is essential for neurons to coordinate how information is processed across the brain. Further studies into this method of communicating information will help to reveal how the primate brain can generate an immense range of behaviors.

difficult to assess these networks. Only small-world topology has been debated (*Yu et al., 2008*; *Gerhard et al., 2011*) and rich-club topology has been shown recently in mice (*Nigam et al., 2016*).

Equally important to topology is the mechanism which coordinates and synchronizes neurons during cognitive or perceptual processes. Previous research has revealed oscillatory synchrony in time as a crucial feature of functional coordination (*Fries, 2009*; *Buzsáki and Wang, 2012*; *Womelsdorf et al., 2014*). Different distinct frequency bands for information transmission and functional network coordination have been identified, such as gamma (40–100 Hz) and theta (4–8 Hz) in the visual areas and up to frontal cortex for coordinated attention selection (*Roelfsema et al., 1997*; *Bosman et al., 2012*; *Gregoriou et al., 2012*), and beta (18–35 Hz) and delta (1–4 Hz) in fronto-parietal regions for network coordination during decision and working memory processes (*Brovelli et al., 2004*; *Pesaran et al., 2008*; *Haegens et al., 2011*; *Salazar et al., 2012*; *Nácher et al., 2013*). Recently, gamma and theta oscillations have been proposed as feedforward communication frequencies across large parts of the visual network, while beta oscillations have been proposed for feedback communication (*Bastos et al., 2015*). However, firing rate correlations have also been found, independent of oscillatory synchronization, to be of importance for communication in the behaving brain (*Fujisawa et al., 2008*; *Smith and Kohn, 2008*). Yet, how functional network topology, described by graph theoretical approaches, relates to oscillatory and non-oscillatory synchronization remains unclear. This question must be answered at the level of single neurons, where oscillatory synchrony can be distinguished from non-oscillatory synchrony.

Here, we recorded in parallel and assessed functional connectivity and network topology from a large number of single neurons (48 to 149 per session) from the primate grasping circuit (*Luppino et al., 1999*), including the ventral premotor (F5), primary motor (M1), and anterior intraparietal (AIP) cortex of three behaving macaque monkeys. Across the three cortical areas we found modular, small-world topology with a clear presence of hubs that were organized as a rich-club. Moreover, rich-club hub neurons predominantly spiked and communicated by oscillatory synchrony in the beta and low frequency range, while the remainder of the network predominately communicated by non-oscillatory synchrony, suggesting that oscillatory synchrony is a central coordination mechanism for functional network topology.

## Results

The current study includes 12 recording sessions from three macaque monkeys (M: 3, S: 6 and Z: 3). We recorded from the grasping motor network, including part of the ventral premotor (F5), anterior

**Table 1.** Trial and single unit counts for all datasets. Marked datasets correspond to the displayed example networks in *Figures 3*–*5* and *Figure 3—figure supplements 1* and *2*. Columns 3–6 show the total and area specific number of units recorded. Columns 7–10 show total and area specific number of units of the largest component of the network, which is the basis for all topological analysis.

| Datasets | Trials | Single units total | F5 | M1 | AIP | Single units used | F5 | M1 | AIP |
|---|---|---|---|---|---|---|---|---|---|
| M 1 | 958 | 149 | 48 | 57 | 44 | 148 | 48 | 57 | 43 |
| M 2 * | 900 | 147 | 52 | 58 | 37 | 137 | 50 | 52 | 35 |
| M 3 | 621 | 107 | 49 | 32 | 26 | 79 | 41 | 20 | 18 |
| S 1 | 503 | 86 | 46 | - | 40 | 57 | 28 | - | 29 |
| S 2 | 565 | 76 | 39 | - | 37 | 64 | 30 | - | 34 |
| S 3 | 460 | 76 | 35 | - | 41 | 64 | 28 | - | 36 |
| S 4 | 460 | 82 | 35 | - | 47 | 64 | 26 | - | 38 |
| S 5 * | 557 | 90 | 42 | - | 48 | 78 | 37 | - | 41 |
| S 6 | 374 | 83 | 42 | - | 41 | 47 | 25 | - | 22 |
| Z 1 | 400 | 52 | 29 | - | 23 | 33 | 21 | - | 12 |
| Z 2 | 436 | 48 | 24 | - | 24 | 30 | 17 | - | 13 |
| Z 3 * | 608 | 59 | 30 | - | 29 | 41 | 21 | - | 20 |
| Average | 570.2 | 87.9 | 39.3 | 49 | 36.4 | 70.2 | 31 | 43 | 28.4 |
| SD | 177.4 | 31.2 | 8.5 | 12.0 | 8.5 | 35.8 | 10.3 | 16.4 | 10.5 |

intraparietal (AIP), and additionally from primary motor (M1) cortex area for monkey M (*Schaffelhofer and Scherberger, 2016*) (*Table 1*). To engage the grasping motor network, monkeys performed a visually-cued delayed grasping task in which the monkey grasped a handle with one of two different grasp types (*Michaels et al., 2015*) (*Figure 1A,B*; see Materials and methods). An average number of 570 trials (SD: 177) were recorded in each session.

In each area, recordings were obtained from two floating microelectrode arrays (FMAs), for a total of 64 channels (32 per microarray) per area (*Figure 1B*; see Materials and methods) from which an average of 88 single units (SD: 32) were recorded in parallel. All recorded single units were modulated by the epochs of the task or the grasp types, clearly indicating the behavioral relevance of the performed task to the detected single units (*Figure 1C*). Nevertheless, in agreement with previous findings (*Buzsáki and Mizuseki, 2014*), firing rates of individual units were relatively stable for different behavioral states of the task following an approximate log-normal distribution (*Figure 1—figure supplement 1*).

## Functional connectivity

The functional connectivity between all simultaneously recorded units of the grasping network was estimated by calculating cross-correlation histograms (CCHs) (*Figure 2A*, *Figure 2—figure supplements 1*, *2*; see Materials and methods), one of the few methods also allowing analyses of the frequency domain (*Bastos and Schoffelen, 2016*) (see below). It is important to stress that the functional connections we describe here do not necessarily represent monosynaptic connections, but merely the influence of one unit onto another. For each neuron pairing one single CCH was estimated over all task epochs and grasp types, since we were interested in the general network interaction and not grasp type or time specific modulations of the network. A general problem of all connectivity measures is common drive to the network, such as stimulus- or movement-locked, but not pairwise, correlations, causing an overestimations of connections. We corrected these biases by subtracting surrogate CCHs (*Figure 2—figure supplement 1A*).

Connections indicated by significant peaks or troughs in CCHs were identified by a cluster-based surrogate test (*Maris et al., 2007*) to all CCHs (see Materials and methods), testing against surrogate CCHs. To control the family-wise error for the entire network, false discovery rate (FDR) correction was applied across all significant connections (*Benjamini and Hochberg, 1995*). For later topological analyses of oscillatory synchrony in the network, we applied Fourier transformations

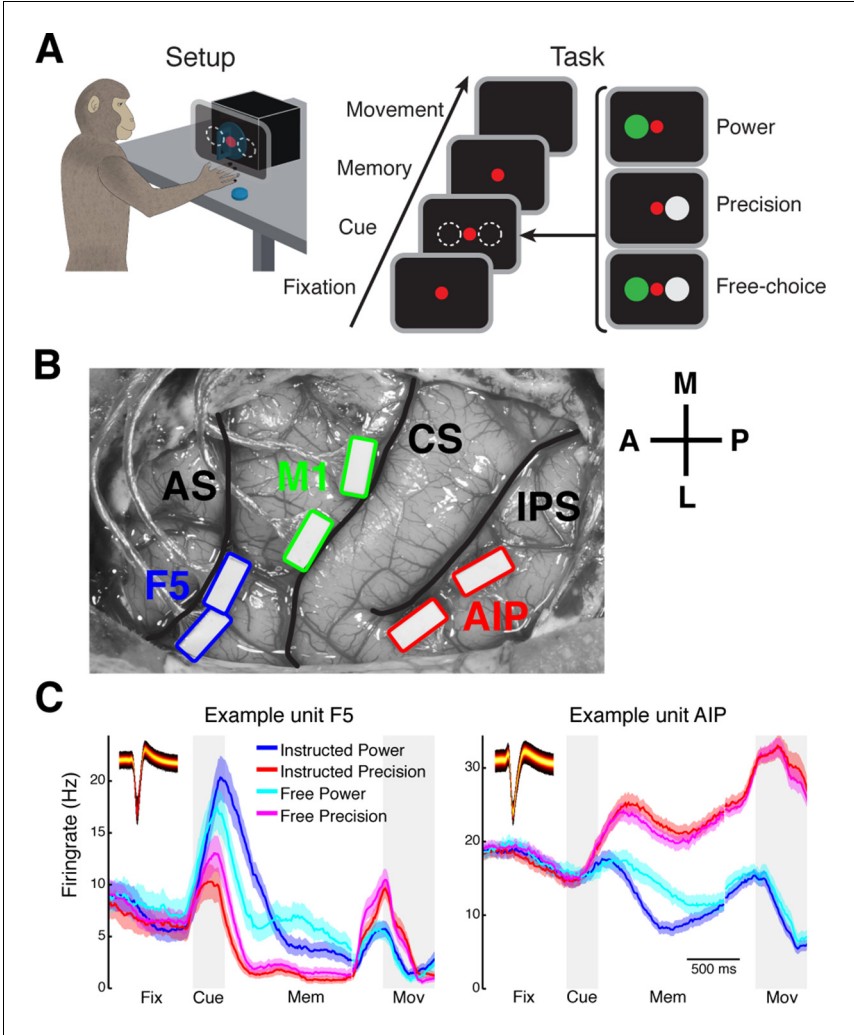

**Figure 1.** Task design and array implantation. (**A**) Choice/no-choice task. Setup: Monkeys were cued to grasp a target (handle) with one of two different grip types displayed on a monitor appearing superimposed on the handle. Task: Monkeys had to fixate a red disk for 600–1000 ms (Fixation), followed by a cue period of 300 ms (Cue). Then, either ('Power') a green disk was presented on the left indicating a power grip, ('Precision') a grey disk on the right indicating precision grip, or ('Free-choice') both disks were presented indicating a free-choice between both grips. After the cue a memory period followed (duration: 1100–1500 ms) before the fixation dot was turned off (go-signal) indicating the monkey to execute the grasp movement (maximum duration:1000 ms). (**B**) Electrode array implantation of monkey M with 6 floating microelectrode arrays (FMAs) in areas AIP, F5, and M1. Arrays were implanted at the lateral end of the intraparietal sulcus (IPS) in AIP, in the posterior bank of the arcuate sulcus (AS) in area F5, and in the anterior bank of the central sulcus (CS) in the hand area of M1. (**C**) Average firing rate across trials of two example units from area F5 (left) and AIP (right). Each colored line corresponds to the mean activity of one condition. Line shadings represent standard error. Inlays shows the corresponding waveforms displayed as density plots.

The following figure supplement is available for figure 1:

**Figure supplement 1.** Firing rate distribution and stability across task epochs and conditions.

(*Figure 2B–D*; see Materials and methods) to all CCHs and auto-correlation histograms (ACHs). The latter detected periodicity in the spiking of individual units, (*Figure 2C*), allowing classifying them as oscillators or non-oscillators.

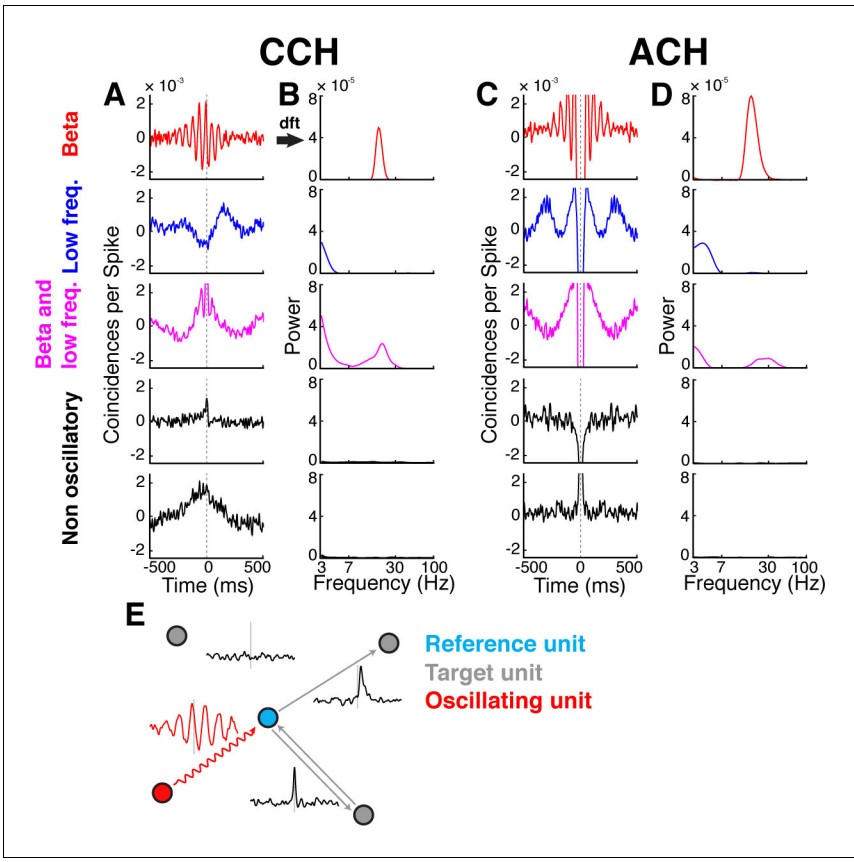

**Figure 2.** Cross- and auto-correlation histograms and frequency spectra. (**A**) Example crosscorrelation histograms (CCHs) for five example neuron pairs. Displayed amplitude is limited to ±2.5x10$^{-3}$ coincidences per spike for better comparison. CCHs are color-coded based on their oscillatory synchronization frequency (red: beta band; blue: low frequencies; magenta: beta and low frequencies; black: no underlying frequency). (**B**) Corresponding frequency spectra of CCHs in a, frequency displayed on logarithmic scale (for better comparison limited to a power of 8x10$^{-5}$) and color-coded as in **A**. (**C**) Same as in **A**, but for auto-correlation histograms (ACHs). (**D**) Same as in **B**, but for the frequency spectra of the ACHs in **C**. (**E**) Illustration of different kinds of CCHs to a reference unit and the inferred connectivity. Upper left: No peak is present in the CCH so the unit is not connected to the reference unit. Upper right: A peak at positive time lags indicates a connection from the reference to the target unit. Lower right: A peak is present straddling the 0 time lag with a maximum peak at 0, indicating a bidirectional connection. Lower left: Several peaks and troughs are present with a clear underlying frequency and a maximum peak at a negative time lag, indicating an oscillatory connection from the target to the reference unit.

The following figure supplements are available for figure 2:

**Figure supplement 1.** CCH processing and statistics, and all connections of an example unit oscillatory synchronized in the low frequency range.

**Figure supplement 2.** All connections of two example units, one non-oscillatory synchronized and one oscillatory synchronized in the beta range.

**Figure supplement 3.** Detectability of directed functional connections using equal rate model simulations.

**Figure supplement 4.** Maximum peak or trough time and phase lag distributions.

---

Directional interaction between pairs of units was inferred from the time delay of significant peaks or troughs in the CCHs (*Figure 2E*). In early studies, a peak or trough in a CCH with a non-zero time lag was classified as a unidirectional connection from one neuron to another while a peak or trough

with a zero time lag was classified as common drive to both neurons (*Moore et al., 1970*). However, recent studies based on complex models rather suggest that zero time lag peaks or troughs in CCHs mainly represent bidirectional connections, which can be explained by the dynamical relaying mechanism, and only rarely reflect a common drive (*Vicente et al., 2008*; *Gollo et al., 2014*). For this reason, we defined zero time lag peaks and troughs in the CCHs as bidirectional connections.

For additional validation of how well we could recover directed functional connectivity, we modeled two sets of 'ground truth' networks with the same distribution of firing rates as recorded single units, one simple network (SN) and one complex network (CN) set (Equal rate model, see Materials and methods). We could detect directed functional connections reasonably well (hits: 62% for SN, and 69% for CN) and hardly detected any false connections (correct rejections (CR) > 99% for SN and CN), independent of the underlying topology (*Figure 2—figure supplement 3B*). To clarify if the missed connections were due to not detecting an existing interaction of a pair of neurons, or due to incorrect classification of directionality, we analyzed the detectability of connections independent of their direction (*Figure 2—figure supplement 3C*), revealing similar results to the detect directed functional connections (hits: 58% for SN, and 69% for CN; CR: > 99% for both). These findings suggest that the missed connections were due to not detecting an existing connection, in accordance with a high accuracy for extracting directionality of only detected connections (*Figure 2—figure supplement 3D*; hits: 97% for SN, and 90% for CN; CR: 75% for SN, and 73% for CN).

Our simulated networks also allowed for a closer evaluation of zero time lag peaks as a result of either common drive or bidirectional connections. In direct comparison, the average common drive CCH as well as the average bidirectional CCH had a maximum at the zero time lag, but with the average bidirectional CCH having a 24 times higher peak (10.89 SD surrogate for bidirectional connections, and 0.45 SD surrogate for common drive; *Figure 2—figure supplement 3E*), which is well in line with around 1% of all common drive pairs were detected as significant. When analyzing the distribution of maximum peaks in more detail, we found more than 7 times more bidirectional connections having a peak at the 0 time lag than common drive pairs (*Figure 2—figure supplement 3F*), in line with the results from the models described above (*Vicente et al., 2008*; *Gollo et al., 2014*). Taken together, all results from the modeled networks show an accurate detectability of directed functional interactions estimated from CCHs.

For a physiological classification of all significantly detected connections, we also analyzed their maximum peak or trough time lag distribution (*Figure 2—figure supplement 4A*). Interestingly, the maximum peak or trough time lag distribution showed an exponential decay, with most of the peaks or troughs having a very short time lag (45.67% < 10 ms, and 85.12% < 100 ms), indicating predominantly direct influences of the units on each other. In case of oscillatory synchronized single units, as strongly present in the data, the classification of the maximum peak or trough time lags was more complex. Given that the maximum peak or trough time lag could be greater than half a cycle of the underlying frequency, it became unclear which unit is leading and which lagging, due to the presence of side lobes (e.g., see *Figure 2A* top panel). Since we found high numbers of oscillatory synchronized single units, predominantly in the beta (20 Hz) and in the low frequency range (4 Hz), as described in detail below, we analyzed the distribution of maximum peaks or troughs phase with respect to the underlying oscillatory frequency (*Figure 2—figure supplement 4B*), and also found an exponential decay, similar to the maximum time lag peak or trough distribution. The majority of phase lags were within half a cycle around the zero time lag for both frequencies (beta connections: 77.70% < $\pi$, low frequency connections: 87.66% < $\pi$), suggesting that for most oscillatory synchronized connections we could accurately determine which unit was leading and which unit was lagging.

For analyzing the functional network topology, all units not connected to the largest inter-connected component were first discarded (mean number of units dropped: 17.75, SD: 9.56; mean percentage: 23.5%, SD: 13.3%; *Table 1*) and binary directional connectivity matrices were created for every dataset (*Figure 3A*). We did not quantify the connection strength, since it has been shown to be biased by different firing rates (*Cohen and Kohn, 2011*).

## Inter-area modular and small-world topology

First, we tested if the networks could be subdivided into modules, such that the number of connections was maximized within and minimized between modules. To properly evaluate modular topology, the fact that connectivity decays with distance has to be considered (*Smith and Kohn, 2008*;

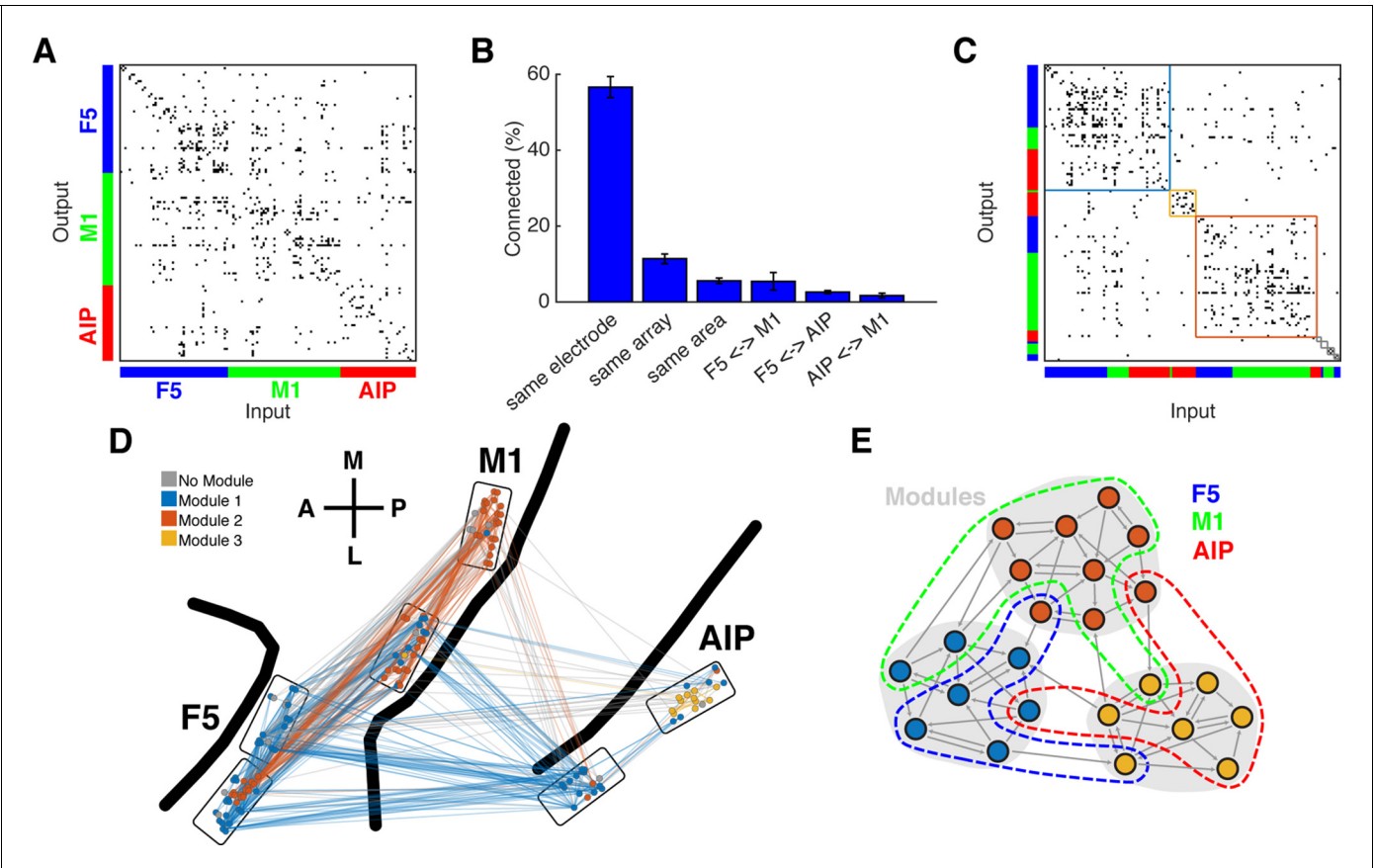

**Figure 3.** Connectivity characteristics and modular topology. (**A**) Connectivity matrix of one dataset from monkey M. Each dot represents a significant connection (Online Methods). Units are ordered by channel number of the recording system. (**B**) Distance dependent connectivity. From left to right: 56,7%, 11,5%, 5,6%, 5,5%,2,6%, and 1,7%. Note the clear distance dependent decay. (**C**) The same matrix as in **A**, but with nodes ordered according to an optimal modularity partition. Colored rectangles surround different network modules. (**D**) Anatomical network representation of the connectivity matrix in **A**. The brain is viewed as in *Figure 1B*. Single units and connections are color coded by module. (**E**) Schematic illustration of modular topology. Modules (dashed regions) consist mainly of single units of one cortical area, but also include small fractions of units from other areas.

The following figure supplements are available for figure 3:

**Figure supplement 1.** Example anatomical networks from Monkey S and Z.

**Figure supplement 2.** Functional network connectivity of an exemplar data set displayed as a web where the locations of all neurons were determined using the visualization of similarities (VOS) approach (*Van Eck and Waltman, 2007*).

*Gerhard et al., 2011*). *Figure 3B* shows the distance-dependent decay of connectivity of our networks according to different subgroups: on the same electrode, on the same array, in the same area, between AIP and F5, between F5 and M1, and between AIP and M1. Connection density was not significantly different within all subgroups (Kruskal-Wallis test, $p > 0.05$).

Modular topology can be quantified by the modularity index Q. If a network can be completely subdivided into modules, Q will be 1. In contrast, if there is no modular structure present at all, Q will be close to 0. We found significant modular topology present in most of the networks (Mean Q: 0.405, SD: 0.087; permutation test, $p < 0.05$, sig. 10/12 datasets), taking the distance-dependent decay of connectivity into account. Modules were significantly predominated by units from a single area (mean largest proportion: 81.4%, SD: 14%; permutation test, $p < 0.001$), but 84% of all modules also included units from other areas, as became apparent when visualized as anatomical networks (*Figure 3D*, and *Figure 3—figure supplement 1A*) or when displayed as a web where the locations of all units are determined by visualization of similarities (VOS) (*Van Eck and Waltman, 2007*)

(*Figure 3—figure supplement 2A,B*). These results reveal a functional modular topology partially not related to the anatomical boundaries between the different areas (*Figure 3E*).

Having shown that a modular topology is present, what is the detailed structure of how individual units are connected within the network? For this, we calculated the cluster coefficient C (with C = 1 corresponding to every neighbor of every unit being interconnected, and C = 0 indicating no interconnections between neighbors) and the average path length, L (defined as the average minimum number of units connecting one unit with another, across all pairs of nodes of the network; see Materials and methods section). If units have dense local clustering (large cluster coefficient C) and can be reached from all other units via a short average path length, L, similar to random networks, the network is considered small-world (SW) (*Watts and Strogatz, 1998*; *Bullmore and Sporns, 2009*). Here, a value of SW >> 1 indicates a small-world topology, whereas SW = 1 corresponds to no small-world effect.

We found significantly higher average cluster coefficients C in comparison to surrogate networks (mean: 0.266, SD: 0.068; permutation test, p<0.001, sig. 12/12 datasets) and on average similar path lengths L (mean: 3.451, SD: 0.823; mean difference to surrogate networks: −0.007; permutation test, p<0.05, sig. higher 5/12, sig. smaller 5/12 datasets). Consequently, all networks had a significant SW-coefficient (mean: 3.05, SD: 0.66; permutation test, p<0.001, sig. 12/12 datasets), suggesting that despite a modular structure the neuronal network is efficiently processing and transmitting information (*Watts and Strogatz, 1998*).

## Degree centrality, betweenness centrality, and hubs

Some networks, have been shown to exhibit heavy-tailed centrality distributions, with a small number of nodes strongly embedded in the network (hubs), which make a strong contribution to the network function (*van den Heuvel and Sporns, 2013a*). A simple and robust measure of centrality is degree centrality (k), which is the number of connections per unit. On average 6.27% (SD: 2.29%) of all possible connections were realized. The degree distribution (*Figure 4A*) was heavy-tailed and best described by an exponential truncated power law model ($P(k) \sim k^{\gamma-1} e^{k/kc}$, $\gamma$ = 0.6839; cutoff degree of kc = 8.657; EXPTPL: adjusted $R^2$ = 0.9891, including a penalty for number of fitted variables), compared to a power law ($P(k) \sim k^{-\gamma}$; PL: adjusted $R^2$ = 0.9177), exponential (EXP: adjusted $R^2$ = 0.9742), or Gaussian (GAUS: adjusted $R^2$ = 0.6826) model. In contrast, surrogate networks with the same distance-dependent connectivity were not heavy-tailed and were best described by a GAUS model (GAUS: adjusted $R^2$ = 0.9655; PL: adjusted $R^2$ = 0.3061; EXPTPL: adjusted $R^2$ = 0.5006; EXP: adjusted $R^2$ = 0.6419). In agreement with the EXPTPL model, networks had significantly more single units within the low, less within the intermediate, and especially more in the high degree range, than surrogate networks (cluster-based permutation test, p<0.05), clear evidence of hubs, independent of distance-dependent connectivity.

A more global aspect of centrality is captured by betweenness centrality (g), an index of the number of shortest paths from all single units to all others that pass through that single unit, normalized by the number of all shortest paths (*van den Heuvel and Sporns, 2013a*). Similar to degree centrality, the betweenness centrality distribution (*Figure 4B*) was heavy-tailed and best described by a PL model, with an estimated exponent of $\gamma$ = 2.212 (PL: adjusted $R^2$ = 0.9753; EXPTPL: adjusted $R^2$ = 0.9745; EXP: adjusted $R^2$ = 0.9593; GAUS: adjusted $R^2$ = −0.1509). The betweenness centrality distribution of surrogate networks was also heavy-tailed and was best described by an EXPTPL model (EXPTPL: adjusted $R^2$ = 0.99; PL: adjusted $R^2$ = 0.9771; EXP: adjusted $R^2$ = 0.9061; GAUS: $R^2$ = −0.5511). Still, in contrast to the PL model, the EXPTPL model had smaller values in the high and low betweenness centrality range. Statistically networks showed a significantly higher number of single units in the low and fewer units in the intermediate betweenness range than surrogate networks (cluster-based permutation test, p<0.05). These findings confirm the presence of hub neurons for betweenness centrality. Units with high degree centrality also tended to have high betweenness centrality (r = 0.75, p<0.001, Spearman correlation), suggesting a coherent group of hub units. We found no significant differences in number of hubs per area (normalized k ≥ 9, g ≥ 0.03; Tukey's honest significant difference test on average group ranks, p<0.05), indicating a distributed hub topology with no area acting as a network center. Together, we have shown that centrality of single units is strongly heterogeneous in the network, with a large group of units being marginally involved in the network and a small group of spatial distributed hub units being extremely central. The presence of hubs provides further evidence of a complex network topology at the single unit level.

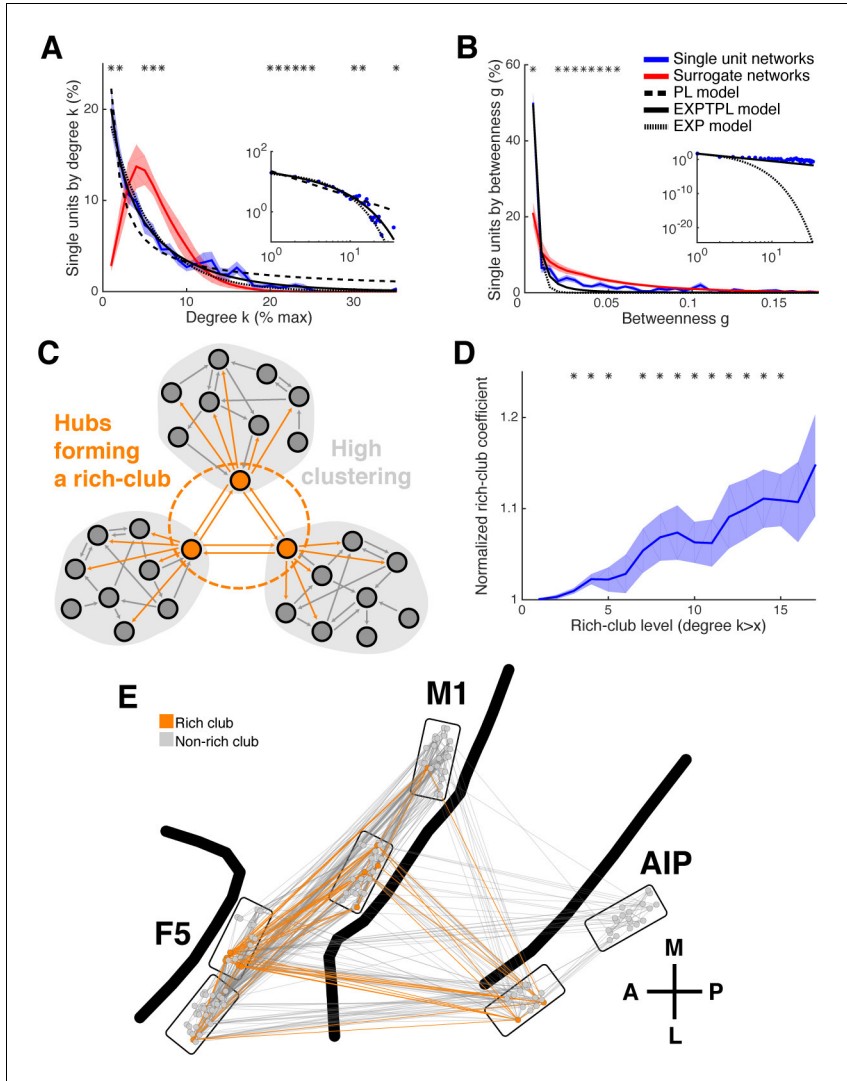

**Figure 4.** Centrality measures, hubs, and rich-club topology. (A) Average degree centrality distribution of all networks (blue) and corresponding surrogate networks (red). Black lines reflect different models fitted to the data (see legend in B). The degree distribution of each dataset was normalized to the possible maximum number of connections per network. The area under the curve was normalized to 100% before averaging. Line shadings show standard error across datasets. Asterisks represent significant differences to surrogate networks. Inlay shows the same distribution and models on a log-log scale. (B) Same as in A, but for the betweenness centrality distribution. Note that the slopes for the EXPTPL and PL model are identical, since the exponential coefficient of the EXPTPL model was zero. (C) Schematic view of a rich-club topology connecting highly clustered modules. (D) Average rich-club level of all datasets relative to surrogate datasets. Asterisks represent significant differences of rich-club level to surrogate networks. (E) Anatomical network representation, as in *Figure 3D*, with connections and units color-coded based on rich-club membership (orange).

The following figure supplements are available for figure 4:

**Figure supplement 1.** Detectability of the underlying network topology using equal rate model simulations.

**Figure supplement 2.** Subsampling model.

However, it has been shown that detectability of functional connections decreases with lower firing rates (*Cohen and Kohn, 2011*). Since the detected firing rates varied approximately across two orders of magnitude (*Figure 1—figure supplement 1B*), this could lead to an underestimation of

degree for low spiking units and an overrepresentation of high firing units as hubs. Therefore, we performed a careful examination of the influence of firing rates on degree and betweenness centrality based on our equal rate model (see Materials and methods). Two sets of networks were tested, simple networks (SNs) and complex networks (CNs), as mentioned previously. SNs had normally distributed connectivity based on the best fitting Gaussian model for the surrogate network degree centrality distribution, while connectivity for CNs were set to precisely resemble the EXPTPL model for the average degree centrality distribution of the measured networks. CNs additionally had a small-world and rich-club topology, as described in the following section.

Differences in firing rate and any possible biases due to the applied method to estimate directed functional connectivity had no effect on the shape of the degree centrality distribution for both kind of networks (*Figure 4—figure supplement 1A*). The betweenness centrality distribution for CNs was also unchanged and only slightly impaired for the SN (*Figure 4—figure supplement 1B*). Nevertheless, the best fitting model for the betweenness centrality distribution of SNs was in neither case (modeled or detected) a PL, as it were for the measured data and the CNs, suggesting no distorting effect by differences in firing rate and the applied method to estimate directed functional connectivity. Importantly, also the average C, average L, and SW-coefficient were correctly detected for both kind of networks.

It is also possible that subsampling, a natural limitation in electrophysiological recordings, could artificially cause a heavy tailed degree centrality distribution even if the underlying connectivity is random (*Han et al., 2005*; *Gerhard et al., 2011*). We simulated a neuronal layer of 32,000 neurons with the same distance-dependent connectivity density as detected in our data (*Figure 3B*), but with Poisson distributed connectivity (*Figure 4—figure supplement 2A*; see Materials and methods). Subsampling was performed in correspondence with our array configuration down to the number of neurons we recorded for real datasets, showing no change to the shape of the degree distribution (*Figure 4—figure supplement 2B*). Only when we decreased the connection density of the model below the detected connectivity in our data was a false heavy-tailed degree distribution apparent (*Figure 4—figure supplement 2C*), which was highly correlated with the networks breaking apart into unconnected components ($R^2 = 0.93$). Additionally, this effect could not be present in our analyzed data since we only analyzed the largest component of the single unit networks. Theses controls suggest that the existence of hubs can neither be explained by distance-dependent connectivity, differences in firing rates, or subsampling.

## Rich-club topology

In some networks hubs exhibit a strong tendency to link to each other, forming a rich-club (*Colizza et al., 2006*), which can be measured by a rich-club coefficient that expresses the tendency of highly connected hub nodes to show above-random levels of interconnectivity (*Figure 4C*). Hub units showed a significantly higher level of interconnectivity than surrogate networks, with up to 15% more connections (*Figure 4D*; cluster-based permutation test, p<0.05).

For our equal rate model, we tested if differences in firing rate and the applied method to estimate directed functional connectivity could cause a false rich-club effect. The present rich-club topology of CNs could be correctly detected, as well as no false rich-club topology was detected for SNs (*Figure 4—figure supplement 1C*). Although the slope of the rich-club coefficient was changed for CNs, rich-club topology was only significant if present (cluster-based permutation test, p<0.05), suggesting a correct representation of rich-club topology for the measured networks.

The rich-club contained neurons from all areas with a rich-club level set to k $\geq$ 9% (*Figure 4E*, *Figure 3—figure supplement 1B* and *2C*; mean rich-club neurons: 27%, SD: 18%; similar results with k set to other levels). A rich-club that spans multiple areas, as described here, has been proposed as a robust structure facilitating efficient communication (*van den Heuvel and Sporns, 2013a*).

## Network topology of oscillatory synchrony

Oscillatory synchronization has been proposed as a mechanism for efficient communication (*Fries, 2009*). As demonstrated above, oscillatory and non-oscillatory synchronized spike patterns for communication could be identified (*Figure 2*, *Figure 2—figure supplement 1B*, *2*). We therefore investigated if specific relationships between distinct frequencies and network topology emerged. Frequency spectra of ACHs of all units and of CCHs that had a significant connection were tested

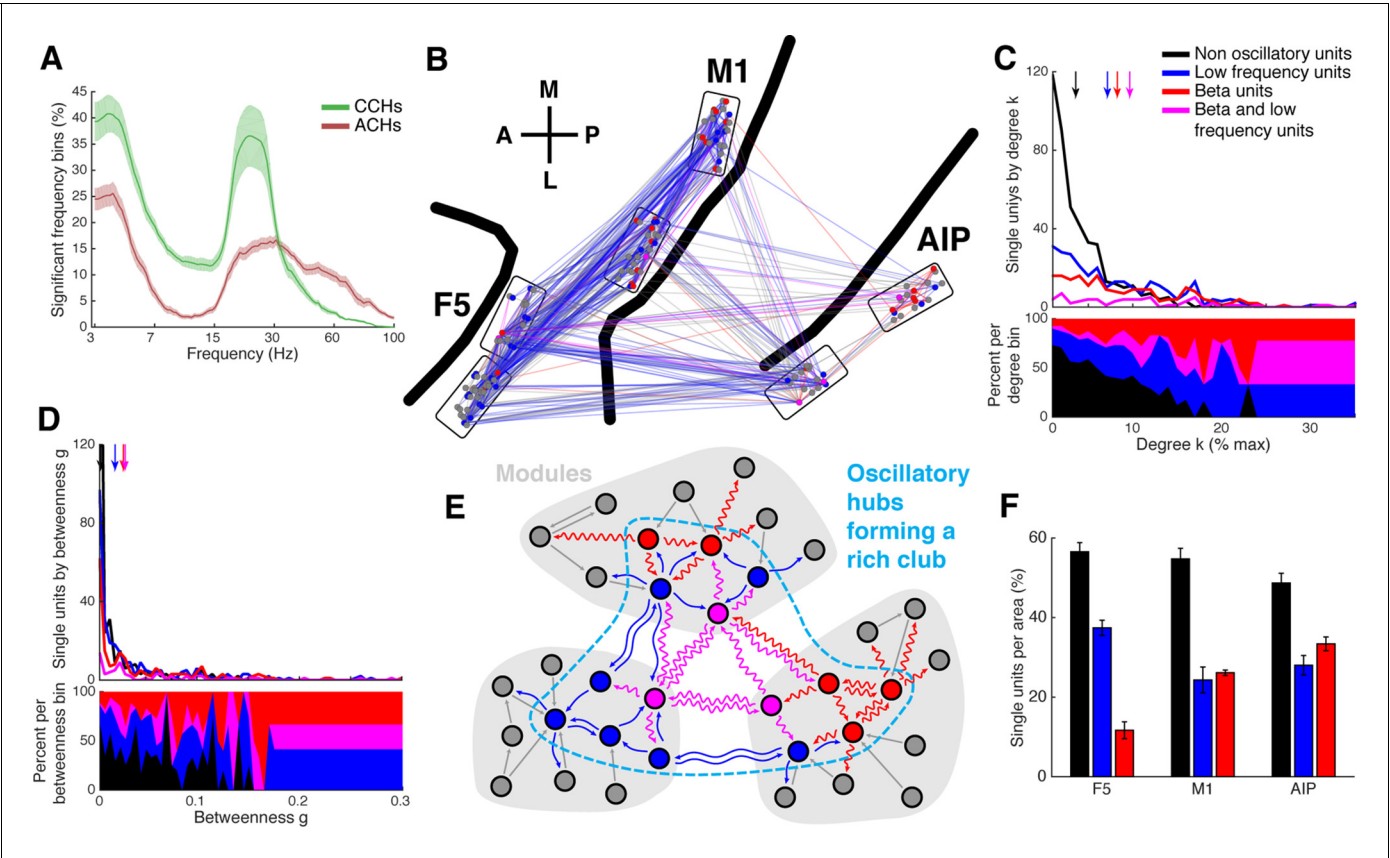

**Figure 5.** Low frequency and beta oscillators and their network topology. (**A**) Average number of significant frequency bins of all ACHs and CCHs over all datasets. Frequencies displayed on a logarithmic scale. Line shadings bars represent standard error across datasets. (**B**) Anatomical network representation as in *Figure 3D* with connections and units color-coded by underlying oscillations (see legend in **C**). (**C**) Degree centrality distribution of all datasets separately for beta and low frequency oscillators, non-oscillators, and single units oscillating in both frequency ranges. Upper panel, summed degree centrality distribution of all single units. Median degree is represented by arrows in corresponding color: beta units: 7.5, low frequency units: 6.3, beta and low frequency units: 8.9, and for non-oscillators: 2.7. (**D**) Same as in **C** but for the betweenness centrality distribution. Median for beta units: 0.023, low frequency units: 0.016, beta and low frequency units: 0.026, and for non-oscillators: 0.001. (**E**) Schematic view of the found network topology of oscillators. Oscillators form a rich-club spanning all areas. (**F**) Distribution of oscillators across areas. The number of single units is normalized to 100% per area. F5 has significantly less beta (red) and significantly more low frequency oscillators (blue) than M1 and AIP. Note that units oscillating in both frequency ranges are counted in both. Non-oscillators (black) still remain the largest group in all areas.

The following figure supplements are available for figure 5:

**Figure supplement 1.** Frequency dependent Hanning windows used for discrete Fourier transform.

**Figure supplement 2.** Sensitivity of CCHs in detecting oscillatory synchrony and non-oscillatory synchrony.

**Figure supplement 3.** Differences in degree centrality and rich-club level for high and low oscillatory state.

for significant frequency bins above chance (cluster-based surrogate test, p<0.05). We found beta (18–35 Hz) and low frequency (3–7 Hz) oscillations predominantly present in the spiking patterns of all datasets (*Figure 5A*, and *Figure 5—figure supplement 1C–E*). Oscillatory synchrony in both frequency ranges was present more often in CCHs (mean beta: 38.3%, low: 44.3%) than in ACHs (mean beta: 22.5%, low: 31.7%), suggesting that the group of oscillating single units (oscillators; *Table 2*) communicates in their underlying frequency to a larger group of units.

Interestingly, there was also a significant group of oscillating single units present in the gamma range (45–80 Hz), which was not mirrored in the CCHs. One possible explanation could be that that

**Table 2.** Number of oscillators in all networks analyzed. Marked datasets correspond to the displayed example networks in *Figure 5* and *Figure 3—figure supplements 1* and *2*.

| Datasets | Oscillators total | Non-Oscillators | Beta Oscillators | Low Frequency oscillators | Oscillators in both frequency ranges |
|---|---|---|---|---|---|
| M 1 | 83 | 65 | 37 | 60 | 14 |
| M 2 * | 60 | 77 | 28 | 37 | 5 |
| M 3 | 34 | 45 | 12 | 25 | 3 |
| S 1 | 31 | 26 | 14 | 26 | 9 |
| S 2 | 32 | 32 | 14 | 22 | 4 |
| S 3 | 31 | 33 | 15 | 20 | 4 |
| S 4 | 26 | 38 | 14 | 19 | 7 |
| S 5 * | 40 | 38 | 22 | 25 | 7 |
| S 6 | 21 | 26 | 14 | 10 | 3 |
| Z 1 | 13 | 20 | 5 | 10 | 2 |
| Z 2 | 13 | 17 | 6 | 9 | 2 |
| Z 3 * | 18 | 23 | 10 | 11 | 3 |
| Average | 33.5 | 36.7 | 15.9 | 22.8 | 5.3 |
| SD | 19.4 | 17.4 | 8.7 | 13.8 | 3.4 |

these units communicate via long-range gamma synchronization with topographically distant areas we did not record, such as the visual cortex (*Gregoriou et al., 2009*).

Oscillators and oscillatory connections were widely distributed and seemed to be very central across all areas (*Figure 5B*, *Figure 3—figure supplement 1C* and *2D*), giving rise to the idea that oscillators could be the hubs of the networks.

*Figure 5C* shows the average degree centrality distribution for all networks, as in *Figure 4A*, but separately for beta and low frequency oscillators, non-oscillators, and units oscillating in both frequencies. There was a clear dominance (high percentage) of oscillators in the high degree range, whereas non-oscillators dominated in the low degree range. The degrees of all three oscillator groups were significantly higher than for non-oscillators (Tukey-Kramer test for rank, p<0.001). Betweenness centrality was also significantly higher for oscillators than for non-oscillators, similar to degree centrality (*Figure 5D*; Tukey-Kramer test for rank, p<0.001). The number of units oscillating in both frequencies was not higher than expected by coincidental overlap of the two frequency bands (permutation test, p>0.05).

Nevertheless, it could be possible that CCHs are more sensitive to oscillatory synchrony than to non-oscillatory synchrony, which would induce a bias when comparing these two groups. At this point, it is important to emphasize that we first tested for significant connectivity independent of oscillatory behavior and only in a second step these connections were tested for their oscillatory behavior as described in the Materials and methods section. This ensured that any detected connection is based on a significant amount (or suppression) of coincidental spikes without any selective sensitivity for oscillatory coupling. As an additional test, we simulated pairs of neurons either with an oscillatory or non-oscillatory firing pattern (see Materials and methods). Since peaks and troughs in CCHs reflect a systematic time lag in spiking between units across trials we simulated different degrees of coupling strengths by systematically varying the trial-wise time offset in spiking for both firing pattern types. Synchronization strength was simply a function of the variation in spike timing offsets between the two neurons and not whether the firing pattern was oscillatory or not (*Figure 5— figure supplement 2*), confirming that oscillatory coupling is not a priori more detectable than non-oscillatory coupling.

Besides these methodological issues already addressed, it is possible that higher firing rates introduce a bias in the statistical detection of significant frequency bins, To control for this possibility, we applied thresholds for the detection of beta and low frequency oscillations. Thresholds were chosen to give, as closely as possible, the same number of beta and low frequency oscillators as statistical

methods. Using this method all three groups had a higher degree and betweenness centrality than non-oscillators, similar to statistical detection (Tukey-Kramer test for rank, p<0.001). To rule out that firing rate dependent detectability of functional connections could cause a spurious inter-dependence of high centrality and detection of oscillatory synchrony, we repeated testing for differences in centrality only with units having a firing rate of 10 Hz and above, confirming that oscillators had significantly higher centrality values (Tukey-Kramer test for rank, p<0.001). Similar results were obtained when we tested the data of each monkeys individually (Tukey-Kramer test for rank, p<0.01). To our knowledge, the current results represent the first evidence that oscillators have a higher centrality in the single unit network than non-oscillators. Consequently, the rich-club of all networks overlapped significantly with oscillating single units (permutation test, p<0.05), highlighting oscillators as the backbone (*van den Heuvel et al., 2012*) of single unit functional connectivity (*Figure 5E*).

The number of oscillators did not differ between areas (Tukey-Kramer test for rank, p<0.05), in agreement with the distribution of hubs as well as rich-club units across areas. Closer examination of oscillator types revealed significantly more beta oscillators in AIP and M1 than in F5, and more low frequency oscillators in F5 than in M1 and AIP (*Figure 5F*; Tukey-Kramer test for rank, p<0.05), reinforcing the notion that different cortical areas operate more strongly in some frequency ranges than others (*Brovelli et al., 2004*).

A further unresolved question is whether a direct relationship exists between oscillatory synchronization and functional rich-club topology. It is well known that oscillatory synchrony in frontal and motor areas appears in short bursts of only a couple of cycles with variable length and amplitude (*Murthy and Fetz, 1996*; *Lundqvist et al., 2016*). We used this property of oscillatory synchrony to split up our data into two equal blocks with high oscillatory and low oscillatory synchrony to investigate the effect on rich-club topology. Since a minimum number of trials are required to properly estimate the functional connectivity for topological analyses, we used the two datasets from monkey M were we recorded more than 900 trials (*Table 1*). The data was split into two blocks with equal number of trials per condition to prevent any biases by different epochs or conditions. Instead of calculating unit-wise ACHs we pooled the activity of all units and estimated single trial population ACHs spectra, reflecting the trial-wise level of oscillatory synchronization. Single trial population ACHs calculations and frequency analyses were performed the same way as for single unit ACHs (see Materials and methods) and divided by their average power in the beta (18–35 Hz) and low frequency (3–7 Hz) band (*Figure 5—figure supplement 1C*). After separation into two blocks, the estimation of functional connectivity and network topological analyses were repeated as if they were two separate datasets. For a valid statement about changes in rich-club topology, the network

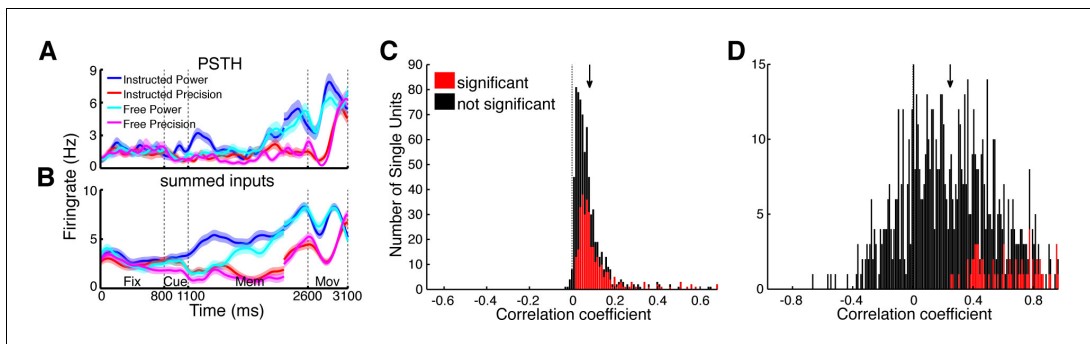

**Figure 6.** Prediction of firing rates based on network topology. (**A**) Average firing rate of one example single unit recorded in F5 in monkey S for the four conditions used in this study during the fixation (Fix), cue (Cue), memory (Mem), and movement period (Mov). The complex tuning patterns for the different task conditions (grip types; free-choice vs. instructed trials) are clearly visible. (**B**) Predicted firing rate of the same unit as in **A** based on the population activity of the connected neurons. Curves in (**A**–**B**) were smoothed with an additional Gaussian kernel (SD: 40 ms). (**C**) Histogram of correlation coefficients between the true and predicted spike trains of all single units of all datasets. Significant correlations are marked in red. Note that hardly any correlation coefficient were negative. (**D**) Histogram of correlation coefficients of condition averaged firing rates. Coloring as in **C**.

structure and in particular the degree distribution, should not be changed. For both datasets the unit-wise degree as well as the degree distribution were very similar (*Figure 5—figure supplement 3B,C*), as well as the betweenness centrality distribution (data not shown). However, when comparing the rich-club level there was a striking difference for higher rich-club levels (*Figure 5—figure supplement 3D*). In both datasets, the high oscillatory state network showed a clear rich-club topology, whereas the low oscillatory state network hardly showed any rich-club effect. These results suggest that a rich-cub topology is only present when there is a high level of oscillatory synchrony in the network.

## Functional network topology and firing rate prediction

Utilizing the identified network topology, the firing rate of individual units can be predicted by the firing rate of input units, providing an estimate on how much of the single unit activity can be explained by functional network connectivity. Each CCH can be understood as a transfer function of spike rates between two units, describing the coincidences per spike at every time point relative to each other. Negative time bins bin reflect input from the reference unit to the target unit while positive time bins reflect the output. To predict the firing rate of a unit, we convolved the spike trains of all units having a significant connection to the corresponding unit with their respective CCHs (output part). Assuming single units to be simple linear integrators, we summed up the individual convolved spike trains (*Figure 6A,B*) and correlated these estimated signals with the original spike trains of the target units smoothed with a Gaussian kernel (SD: 3.66 ms), identical to the CCH smoothing. Ninety-nine percent of predicted firing rate curves were positively correlated with the real firing rates of the corresponding target units (*Figure 6C*).

However, these correlations could also be due to synchronous up and down states of the brain (*Gilbert and Sigman, 2007*), which makes proper statistical testing obligatory. Three different permutation tests were applied: shuffling of trials, shuffling of the output parts of CCHs, and shuffling of input units. Only if the correlation coefficient significantly exceeded all three permutation distributions ($p < 0.05$), the correlation was considered significant. Remarkably, 45% of the firing rate patterns of our single units could be significantly predicted by their inputs. The differences between grasp types and decision conditions could be significantly predicted in 9% of all cases (*Figure 6D*; positive correlation: 79%; shuffling of the transfer kernels and input units, $p < 0.05$), even using this simple approach that involved no parameter fitting. The functional network topology presented here allows a surprisingly accurate prediction of temporal firing dynamics, suggesting that the network captured in our recordings, despite being a small subset of the entire network, accurately represents a large portion of the relevant communication in the fronto-parietal grasping network.

## Discussion

We analyzed single unit functional network topology across several cortical regions of three monkeys performing a delayed grasping task. The network was structured as a complex network (*Bullmore and Sporns, 2009*) with a modular SW topology, and highly central hub-units localized in all three areas forming a rich-club. The advantage of such a topology is that it allows for fast and dynamical information processing combined with high robustness against errors (*Barabási and Oltvai, 2004*; *Bassett and Bullmore, 2006*; *Bullmore and Sporns, 2009*; *van den Heuvel et al., 2012*). More detailed analyses of the kind of synchronization processes within the network revealed that the population of single units could be divided into two groups: oscillatory spiking and synchronized units in the low frequency range or in the beta range, and a group of non-oscillatory spiking units. Importantly, the hubs and therefore the rich-club consisted predominantly of oscillators, while the peripheral neurons were predominantly non-oscillators.

Why is oscillatory synchrony such a central element of functional network topology? More and more evidence supports the hypothesis that information is propagated not only as a simple rate code, but by feed-forward coincidence detection accomplished by oscillatory synchrony (*Fries, 2009*), meaning that phase-synchronization of neurons with one another is used as a selection mechanism for information transmission. The advantage of this mechanism is not only a reduction of energy cost, but also rhythmic gain modulation. By changing the phase of a synchronous neural population, such as in high-order areas, the input of one group of neurons can be selectively amplified as inputs to another group of neurons, allowing for high selectivity and high flexibility, which are

exactly the requirements a hub has to fulfill (*van den Heuvel and Sporns, 2013a*). While feed-forward coincidence detection can theoretically also be accomplished by non-oscillatory processes (*Fries, 2009*), the coordination of a network spanning different areas requires a larger group of neurons to fire in a coherent manner (*Buzsáki and Wang, 2012*). A rich-club of oscillating neurons is exactly that, a coherent structure cross-linking functionally segregated modules (*van den Heuvel and Sporns, 2013b*), suggesting oscillators act as a backbone promoting and coordinating functional communication across different cortical areas (*van den Heuvel et al., 2012*). This hypothesis is also in accordance with the finding that synchronization over larger distances (>2 mm) is almost always oscillatory, whereas synchronization over short distances occurs also in the absence of oscillations (*König et al., 1995*).

What are the roles of the two different distinct frequency bands present in this network? Parietal and motor areas have been found to communicate via ~20 Hz beta synchronization (*Pesaran et al., 2002*; *Brovelli et al., 2004*; *Pesaran et al., 2008*; *Dean et al., 2012*) and an increment in beta band activity seems related to the maintenance of the current sensorimotor or cognitive state, in agreement with findings in the basal ganglia (*Engel and Fries, 2010*). Oscillatory synchrony in the low frequency range (1–4 Hz) has been shown to be important for communication within and between the prefrontal and motor areas (*Siegel et al., 2009*; *Nácher et al., 2013*) and as a potential population mechanism of movement generation in motor and premotor cortex during reach initiation (*Churchland et al., 2012*). Therefore, beta seems to be a stabilizing signal, low frequencies a global coordination signal, and both are involved in movement initiation with opposing roles. One possibility is that a function of the rich-club, composed of beta and low frequency oscillators spanning parietal and prefrontal cortex, is coordinating movement generation and initiation. Another possible explanation is that the power of fast oscillations is modulated by the phase of slow oscillations, termed cross-frequency phase-amplitude coupling, which could serve as a neuronal syntax for information transmission (*Buzsáki, 2010*; *Buzsáki and Mizuseki, 2014*). Our observation of oscillators in both frequency ranges simultaneously (third row of *Figure 2C,D*, and *Figure 5C,D*) supports this concept.

Interestingly, we found that beta oscillators were present most frequently in AIP, followed by M1, hardly in F5, and in reverse order for low frequency oscillators (*Figure 5F*). This is in line with the previous findings that information via beta band is primarily transmitted from the parietal to the frontal regions and not vice versa (*Brovelli et al., 2004*). In areas that are hierarchically lower than the parietal lobe, such as the visual system, beta was identified as a top-down communication frequency (*Bastos et al., 2015*). Therefore, the parietal lobe might be a center of beta generation. Low frequency oscillatory synchrony during active behavior has been found predominantly in prefrontal areas (*Siegel et al., 2009*; *Nácher et al., 2013*). We speculate that the center of low frequency oscillation could be in the prefrontal cortex, suggesting that different anatomical regions generate and communicate with different frequencies. The exact reason for the presence of distinct frequency bands for communication and their detailed interplay needs to be addressed in future studies.

The single unit network topology was highly similar to the regional network of the brain measured by EEG, MEG, DTI or fMRI (*Bullmore and Sporns, 2009*; *Rubinov and Sporns, 2010*; *van den Heuvel et al., 2012*; *van den Heuvel and Sporns, 2013a*), which strongly suggests that the observed topological properties are scale-invariant (*Bullmore and Sporns, 2009*). Oscillatory synchrony may therefore act as a global coordination mechanism across the whole cortex.

The modules of the network were primarily composed of the individual areas themselves. Yet, most modules also consisted of a small, but significant, proportion of units from other areas, indicating that the anatomical distance does not necessarily reflect the functional distance. This finding is in line with a recent study showing that the population of neurons within one area can be split up into 'choristers,' which are strongly coupled to the rate of the whole population, and 'soloists,' which are not (*Okun et al., 2015*). We speculate that 'soloists' could be part of functional circuits centered in other brain areas, in accordance with the present modular topology.

Since we recorded only from a subpopulation of the actual network, it was important to evaluate whether the observed network topology sufficiently represented the fronto-parietal grasping network. We demonstrated that a significant amount of the firing rate of single units could be predicted using only their network inputs, even for complex tuning patterns, suggesting that even a small fraction of the network is enough to characterize a reasonable amount of the spatio-temporal spiking dynamics. Furthermore, we demonstrated on a model that subsampling from a huge network with

the same distance-dependent connectivity density as detected in our data did not affect the shape of the degree distribution (*Figure 4—figure supplement 2*). For these reasons, we are confident that our analyzed single unit network constitutes a significant representation of the underlying network dynamics.

One possible point of misinterpretation of the functional network structure could be common drive, resulting in an overestimation of connectivity. Our method to detect functional connectivity corrects for common drive due to stimulus- and movement-locked inputs as well as for trial-wise fluctuations in spiking. Nevertheless, there are two possible additional sources of common drive. The first is the possibility that two neurons receive input from a third neuron while themselves being functionally uncoupled, resulting in a significant peak in the CCHs due to their input similarity. We investigated this possibility using our equal rate model, which included physiologically plausible firing rates and pairwise correlations. Common drive pairs of simulated simple or complex networks were detected as being significant in only around 1% of all cases, suggesting that, irrespective of the underlying topology, our method for detecting functional connectivity is hardly biased by pairwise common drive. The second possibility is that cortical columns or areas could receive common drive input that would cause these neurons to fire in a synchronized fashion even if they were functionally uncoupled. In such a scenario two things would be expected: first, units on the same electrode, as well as units in the same area, should show a similar connectivity pattern. Second, all neurons in the network should show a similar number of functional connections, since they are synchronized by common drive, resulting in a uniform degree centrality distribution. However, we found 43% of all neurons on the same electrode to be not connected, and only sparse connectivity was found in the same area with strongly connected pairs of neurons next to unconnected pairs (e.g., *Figure 2—figure supplement 1B*, *2*). Most importantly, the degree distribution of the measured networks was highly heterogeneous and heavy-tailed in contradiction to what would be expected by a strong influence of column- or area-specific common drive. Therefore, it is unlikely that event unrelated common drive can account for a significant amount of the detected functional connections. Further evidence arises from the fact that we found beta, low frequency, and non-oscillatory synchronization with different maximum peak or trough time time and phase lags (*Figure 2—figure supplement 3*), present simultaneously across all areas, also not consistent with a global common drive bias.

To our knowledge, these results provide the first evidence of oscillatory synchrony as a central coordinating mechanism for the formation of functional network topology at the single neuron level. The combination of communication properties of oscillating single units and their functional topology adds an essential dimension to the understanding of neural circuits. By demonstrating that oscillating neurons form a backbone for functional connectivity, spanning several areas, we provide a unified basis for understanding the neuronal computations coordinating and generating behavior at the network level.

## Materials and methods

### Basic procedures

Neural activity was recorded simultaneously from many channels in two female and one male rhesus macaque monkey (Animals S, Z, and M; body weight 9, 7, and 10 kg, respectively). Detailed experimental procedures have been described previously (*Michaels et al., 2015*). All procedures and animal care were in accordance with German and European law and were in agreement with the *Guidelines for the Care and Use of Mammals in Neuroscience and Behavioral Research* (*National Research Council, 2003*).

### Behavioral task

*Figure 1A* illustrates the time course of the behavioral task as described previously (*Michaels et al., 2015*). Trials started after the monkey placed both hands on the resting positions and fixated a red fixation disk (fixation period). After 600 to 1000 ms, cues in the form of disks were shown next to the fixation disk for 300 ms to instruct the monkey about the required grip type (power or precision; cue period). During this epoch the grasp target, a handle, was also illuminated. In the instructed task one disk was shown, while in the free-choice task both disks were turned on, indicating that the

monkey was free to choose between the two grip types. The monkey then had to memorize the instruction for 1100 to 1500 ms (memory period). The switching off of the fixation light cued the monkey to reach and grasp the target (movement period) in order to receive a liquid reward. Importantly, during free choice trials the reward was iteratively reduced every time the monkey repeatedly chose the same grip type. All trials were randomly interleaved and executed in darkness. The behavioral task also contained delayed instructed trials, which were not analyzed in this study.

## Chronic electrode implantation

Surgical procedures have been described previously (*Michaels et al., 2015*). In short, each animal was implanted with two floating microelectrode arrays per area (FMAs; Microprobes for Life Sciences; 32 electrodes; spacing between electrodes: 400 µm; length: 1.5 to 7.1 mm monotonically increasing to target grey matter along the sulcus). Animal S and Z were implanted with four FMAs in area AIP and F5 in the left and the right hemisphere, respectively. Animal M was implanted with a total of six FMAs in the same cortical areas and two additional arrays in area M1, in the left hemisphere (*Figure 1B*).

## Neural recordings and spike sorting

Neural signals from the implanted arrays were amplified and digitally stored using a 128 channel recording system (Cerebus, Blackrock Microsystems; sampling rate 30 kS/s; 0.6–7500 Hz band-pass hardware filter; for monkey S and Z) or a 256 channel Tucker-Davis system (TDT RZ2; sampling rate 24.414 kS/s; 0.6–10,000 Hz band-pass hardware filter; monkey M).

For spike detection, data were first low-pass filtered with a median filter (window length 3 ms) and the result subtracted from the raw signal, corresponding to a nonlinear high-pass filter. Afterwards the signal was low-pass filtered with a non-causal Butterworth filter (5000 Hz; fourth order). To eliminate common noise-sources principal component (PC) artifact cancellation was applied for all electrodes of each array as described previously (*Musial et al., 2002*). To ensure that no individual channels were eliminated, PCs with any coefficient greater than 0.36 (conservatively chosen and with respect to normalized data) were retained. Spike waveforms were detected and semi-automatically sorted using a modified version of the offline spike sorter *Wave_clus* (*Quiroga et al., 2004*; *Kraskov et al., 2009*).

Units were classified as single- or non-single unit based on five criteria: (1), the absence of short (1–2 ms) intervals in the inter-spike interval histogram for single units; (2), the homogeneity and SD of the detected spike waveforms; (3), the separation of waveform clusters in the projection of the first 17 features (a combination for optimal discriminability of PCs, single values of the wavelet decomposition, and samples of spike waveforms) detected by *Wave_clus*; (4), the presence of well-known waveform shapes characteristics for single units; and (5), the shape of the inter-spike interval distribution.

After the semiautomatic sorting process, redetection of the different average waveforms (templates) was done to detect overlaid waveforms (*Gozani and Miller, 1994*). To achieve this, filtered signals were convolved with the templates starting with the biggest waveform. Independently for each template, redetection and resorting was run automatically using a linear discriminate analysis for classification of waveforms. After spike identification, the target template was subtracted from the filtered signal of the corresponding channel to reduce artifacts during the detection of the next template. This procedure allowed us to detect spikes with a temporal overlap up to 0.2 ms. Unit isolation was evaluated again, based on the five criteria mentioned above, to determine the final classification of all units into single or non-single units. Stationarity of firing rate was checked for all units and in case it was not stable over the entire recording session (more than 30% change in firing rate between the first 10 min and the last 10 min of recording) the unit was excluded from further analyses (~3% of all single units). Only single units fulfilling all of these criteria, and no multi-units, were further used in this study.

## Functional connectivity analysis

After sorting, spike events were binned in non-overlapping 1-ms windows to produce a continuous firing rate signal (1 kHz) and aligned to cue and movement onset. Two time windows were chosen for further analysis (Cue onset: −700 to 1500 ms; Movement onset: −300 to 500 ms), since neuronal

activity was locked to both events, with a variable memory period between them. Note that all three monkeys had very consistent movement times (mean SD across datasets = 39 ms).

The functional network topology of single-unit populations was derived from analyses of pairwise correlations (*Yu et al., 2008*). We calculated cross-correlation histograms (CCHs; time lags: −500 ms to 500 ms) between all pairs of single units of each dataset (*Bair et al., 2001*):

$$CCH_{n_1,n_2}(\tau) = \frac{1}{M}\sum_{i=1}^{M}\sum_{t=1}^{N}\frac{x_{n_1}^i(t)\,x_{n_2}^i(t+\tau)}{(N-|\tau|)\,\sqrt{\lambda_1\lambda_2}} \tag{1}$$

where $M$ is the number of trials, $t$ is time, $N$ is the number of time bins in the trial, $x_{n_1}^i$ and $x_{n_2}^i$ are the spike trains of single units $n_1$ and $n_2$ on trial $i$, $\tau$ is the time lag, and $\lambda_1$ and $\lambda_2$ are the mean firing rates of the two single units across the entire time interval $M$. The denominator is normalizing for the degree of overlap $(N-|\tau|)$ in the CCH and the geometric mean spike rate $\sqrt{\lambda_1\lambda_2}$, which is the most common normalization used for CCHs (*Bair et al., 2001*; *Smith and Kohn, 2008*). The normalized CCHs were then averaged across all time periods and task conditions (e.g., see *Figure 2—figure supplement 1A*).

Subsequently, all CCHs were corrected for correlations induced by common stimulus drive or global state changes, such as arm and hand movements, as well as for trial-wise fluctuation in spiking, by simulating and subsequently subtracting surrogate CCHs. Surrogate CCHs contain the same stimulus locked correlation, but no pairwise temporal correlation. To this end, peri-stimulus time histograms (PSTH) were calculated for the same two time windows and alignments (Cue and Movement onset) as mentioned above, separately for each single unit and task condition (smoothed with a Gaussian kernel, SD: 3.66 ms). Artificial spike trains were generated from an inhomogeneous Poisson process using the PSTHs as the rate function (*Ramalingam et al., 2013*). These artificial spike trains preserved the number of trials and the number of spikes per trial, but varied in the timing of individual spikes (surrogate data; e.g., *Figure 2—figure supplement 1A*). Since the number of spikes per trial was preserved for all units recorded simultaneously, any trial-wise common drive is equally present and therefore accounted for in the surrogate data (*Smith and Kohn, 2008*). From these surrogate data, surrogate CCHs were calculated by replacing $x_n^i$ with the trials of the artificial spike trains for the corresponding single unit (surrogate CCHs). This procedure was repeated 1000 times. The resulting surrogate CCHs reflected the level of correlation when both units are statistically independent. Finally, average surrogate CCHs were subtracted from the CCHs to yield the corrected CCHs.

Auto-correlation histograms (ACHs) were generated by setting $x_{n_1}^i = x_{n_2}^i$ in *Equation 1* for all $i$, and corrected by generating artificial spike trains and substituting them for $x_{n_1}^i$ and $x_{n_2}^i$ in *Equation 1* for the calculation of surrogate ACHs.

## Cluster-based surrogate test

For statistical purposes, all surrogate CCHs were corrected by their own average to achieve an equally processed set compared to the corrected CCHs, containing just the chance level of correlation (corrected surrogate CCHs). These 1000 corrected surrogate CCHs were then used to run a nonparametric cluster-based surrogate test, a variation of the cluster-based permutation test (*Maris and Oostenveld, 2007*), to deal with the multiple comparison problem of testing all time lags. Cluster-based tests are tests for dependent variables, which consider contiguous values fulfilling a certain criterion as a cluster. Instead of calculating a test statistic for individual values, the accumulated values of clusters are tested against a null distribution of accumulated cluster values by chance. In our case, adjacent time lags are not independent, since functional coupling of neurons does not follow millisecond precision. We checked significance for a time window of −200 ms to 200 ms. Calculation of this test statistic involved the following steps:

1. For every time bin the standard deviation of corrected surrogate CCHs was calculated. Subsequently, the corrected CCH and the corrected surrogate CCHs were normalized by these standard deviations (z transformation of the data).
2. A z-score of 2 corresponds to a p-value of ~0.05. So we marked all time lags exceeding a z-score of 2 or −2. Please note that the statistical inference is not directly based on this z-score criterion, but rather on the subsequent non-parametric test.

3. As already mentioned, in CCHs neighboring time lags are not independent. Clusters of marked bins were selected on the basis of temporal adjacency.
4. From each corrected surrogate CCH, the largest cluster was selected (independent of the sign) based on its accumulated z-score, creating a distribution of 1000 largest clusters. Since we used each unit as $x^i_{n_1}$ and as $x^i_{n_2}$, we obtained two CCHs per pair of units. These two CCHs are identical, except for being inverted in time. We merged their distributions to a final distribution of the 2000 largest chance clusters.
5. In a final step, cluster-level statistics were calculated. The accumulated z-score of each real cluster was tested against the distribution of biggest clusters occurring by chance. The obtained p-value of each cluster was saved for further corrections.

This procedure was repeated for every CCH. A critical alpha-level of 0.05 was selected. Nevertheless, at this processing step we still have a total alpha-error equal to our set criterion times the number of single unit pairs tested. For complete multiple comparison correction, false discovery rate correction was applied on all found clusters across all compared pairs of single units (*Benjamini and Hochberg, 1995*) to yield

$$P_{(k)} \leq \frac{k}{m}q \qquad (2)$$

where q is our set criterion of 0.05 false positives, m the total number of clusters, k = 1,…,m, and $P_{(k)}$ are the p-values of all clusters in increasing order. All clusters whose p-values did not fulfill *Equation 2* were rejected. By doing so we achieved a total alpha-level of 0.05 for each dataset.

## Network analysis

For every pair of neurons it was evaluated if there were significant troughs or peaks in their CCHs. If there was only a trough or peak with negative (or positive) time lags, this pair was denoted as having a connection from the input to the target (or the target to the input) unit (*Figure 2E*). In case there were several clusters on both sides of the zero time lag, or a cluster straddling the zero time lag, we checked the unsigned maximum peak of the corresponding CCH. If the maximum peak was shifted more than 2 ms to either side, the connection was considered unidirectional, as described before. Otherwise, the connection between the two single units was considered functional bidirectional (*Figure 2E*), since the units are driven by the circuit at the same time. We systematically varied the maximum peak shift (0–5 ms) for bidirectional classification with little to no change to the results. Repeating this procedure for all pairs of single units led to a binary directed connectivity matrix (*Figure 3A*).

To characterize brain networks on every scale, network measures from the multidisciplinary field of graph theory were utilized (*Rubinov and Sporns, 2010*).

A network is defined by the nodes (*N*) and connections between pairs of nodes. In our network nodes represented single units. For all following network measures, *n* is the number of nodes and *l* the number of connections. $a_{ij}$ is the connection between nodes *i* and *j*: $a_{ij} = 1$ if the link $(i, j)$ exists and $a_{ij} = 0$ otherwise ($a_{ii} = 0$ for all *i*). Furthermore, we define:

**Degree centrality, $k_i$,** is the number of connections to a node *i*.

$$k_i = \sum_{j \in N} a_{ij} \qquad (3)$$

**Shortest path length, $d_{i,j}$,** is the minimum number of nodes connecting nodes *i* and

$$d_{ij} = \sum_{a_{uv} \in g^{i \leftrightarrow j}} a_{uv} \qquad (4)$$

*j*. where $g^{i \leftrightarrow j}$ is the shortest path between *i* and *j*.

**Characteristic path length, *L*,** is the average shortest path length between all pairs of nodes of the network.

$$L = \frac{1}{n(n-1)} \sum_{\substack{i,j \in N \\ i \neq j}} d_{ij} \tag{5}$$

**Betweenness centrality,** $g_i$, is the average fraction of shortest paths that pass through node $i$.

$$g_i = \frac{1}{(n-1)(n-2)} \sum_{\substack{h,j \in N \\ h \neq j, h \neq i, j \neq i}} \frac{\rho_{hj}^{(i)}}{\rho_{hj}} \tag{6}$$

where $\rho_{hj}$ is the number of shortest paths between $h$ and $j$, and $\rho_{hj}^{(i)}$ is the number of shortest paths between $h$ and $j$ that pass through $i$.

**Clustering coefficient of the network, C,** is the average fraction of existing to maximal possible interconnections between all directly connected nodes to node $i$.

$$C = \frac{1}{n} \sum_{i \in N} \frac{2t_i}{k_i(k_i - 1)} \tag{7}$$

Where $k_i$ are all connected neighbors to node $i$ and $t_i$ is the number of links between them.

**Small-worldness, SW**, is the ratio of $C$ and $L$ each normalized by the same measurements for a size matched random network.

$$SW = \frac{C/C_{rand}}{L/L_{rand}} \tag{8}$$

Small-world networks are formally defined as networks that are significantly more clustered than random networks, yet have approximately the same characteristic path length as random networks (*Watts and Strogatz, 1998*).

**Modularity, Q**, is the proportion of all links within modules $M$ with links between modules, when the network is fully subdivided into non-overlapping modules in a way that maximizes the number of within-group connections and minimizes the number of between-group connections.

$$Q = \sum_{u \in M} \left[ e_{uu} - \left( \sum_{v \in M} e_{uv} \right)^2 \right] \tag{9}$$

where $e_{uv}$ is the fraction of all links that connect nodes in module $u$ with nodes in module $v$.

**Rich-club coefficient, R**, at degree $k$ is the fraction of connections between all nodes of degree $k$ or higher, with respect to the maximum possible number of such connections.

$$R(k) = \frac{2E_{>k}}{N_{>k}(N_{>k} - 1)} \tag{10}$$

where $E_{>k}$ is the number of connections among the $N_{>k}$ nodes having degree of $k$ or higher (*Colizza et al., 2006*). To reduce inaccuracy for large degrees we calculated the rich-club coefficient only in degree bins containing at least 5 single units ($N_k \geq 5$).

## Statistics for network measures

For statistical purposes we created two types of surrogate network sets per dataset (1000 partitions each). All surrogate networks were created by shuffling the connectivity matrix. Since connectivity is a function of distance (*Smith and Kohn, 2008*; *Gerhard et al., 2011*), distance dependency was reflected in our surrogate data. During shuffling, the number of connections for single units on the same electrode, the same array, the same cortical area, and the different inter-area connections were always held constant (*Figure 3B*). For all surrogate networks, the total number of single units, number of connections, and the distance-dependent ratio of bi- and uni-directional connections were kept as similar as possible to the original connectivity matrix with only the required network parameter shuffled. We used these sets of surrogate networks to test the small-world coefficient,

the degree centrality distribution, and the betweenness centrality distribution. Statistical testing of the rich-club coefficient and conservative testing of modularity requires surrogate networks with a matched degree centrality distribution. To this end, we generated a second set of surrogates networks with the degree distribution preserved. One issue that could arise due to shuffling is that the connectivity matrix of some units or groups of units could become disconnected from the main part of the network, since the calculation of most network measures requires a fully connected, not segregated, network. For this purpose, each surrogate network was tested for segregation into different components. If a network was segregated, it was discarded and the process repeated until 1000 non-segregated networks were generated.

To determine if the degree, the betweenness centrality distribution, or the rich-club level were significantly different to surrogate networks, we used a nonparametric cluster-based permutation test (*Maris and Oostenveld, 2007*). Briefly, this test evaluates the t-statistic (independent samples) between centrality or rich-club distributions and their surrogate distributions over all data points exceeding a critical alpha-level set to 0.05. In a second step, adjacent degree, betweenness values, or rich-club coefficients exceeding the set alpha-level are considered as clusters, extracted, and their t-value summed. A test distribution was generated by randomly permuting the centrality or rich-club distributions across recording days and monkeys with the corresponding surrogate distributions by randomly reassigning them to one of the two groups while maintaining the group size. For each partition (1000 partitions) the t-statistics and clustering was repeated. From every partition the largest cluster-level statistic was used to generate a largest chance cluster distribution. For each real cluster-level statistic a nonparametric statistical test was performed by calculating a p-value under the largest chance cluster distribution. Thus, the multiple comparisons for each sample are replaced by a single comparison, replacing the need to make multiple comparisons.

Since some electrode pairs between F5 and M1 are closer than some other pairs within M1 for monkey M, we repeated statistics for network measures for all datasets from monkey M with physical distance dependent shuffling instead of the above mentioned categories such as 'same electrode', 'same array,' and 'same area'. To this end, we calculated the pairwise physical distance between all pairs of electrodes based on an anatomical diagram (*Figure 1B*) and defined distance groups with a stepsize of 3.6 mm including 0 mm as one group. The physical distance between AIP and the two other areas is misleading, since the neuronal axons have to pass the central sulcus. Therefore, we set all distances between AIP and the two other areas as a separate maximum distance group. Note that we had to define groups to be able to shuffle connections. Nevertheless, the categorical distance dependent shuffling was subdivided into 8 groups, which is more conservative than the 6 groups defined in the original analysis. All statistics for network measures gave nearly identical results, with no case where a measure was significant when it was not for categorical distance dependent shuffling, and vice versa for non-significant measures. In addition, the normalized rich-club coefficient, which depends on the surrogate networks, was highly correlated (r = 0.98) between the two different ways of distance dependent shuffling.

## Equal rate model

For validation of the estimates of directed functional connectivity, as well as to check for a possible bias in the detected network topology obtained using CCHs, we modeled artificial directed neuronal networks with the same firing rate distribution as the recorded single units. Two sets of networks were generated, one simple network (SN) set with normally distributed connectivity and one complex network (CN) set with heterogeneously distributed connectivity, and in agreement with previous studies both with weak connection strength between neuronal pairs (*Cohen and Kohn, 2011*).

For each simulated neuron, artificial spike trains were generated with Poisson distributed firing and an average rate randomly drawn from the real firing rate distribution. For the SN set, the number of connections from each neuron to other neurons was drawn randomly from a Gaussian distribution (mean: 5.22, SD: 3.214), mirroring the average degree centrality distribution of surrogate networks. For the complex network set (CN), the number of connections followed precisely the EXPTPL model for the average degree centrality distribution of the measured networks (*Figure 4A*), with a weak rich-club and small-world topology. In case one neuron was connected to another, spikes were added in a probabilistic manner for a certain amount of time, starting with time point $t + 1$ in ms relative to the spike event, reflecting the axonal delay. The network was updated every

millisecond, allowing for multiple interactions. Gamma functions were used as temporal transfer kernels, given by

$$f(t|a,b) = \frac{1}{b^a \Gamma(a)} t^{a-1} e^{\frac{-t}{b}} \tag{11}$$

where $f$ is the probability of an additional spike appearing, $t$ is time in ms, $a$ is a constant set to 5 and $b$ is randomly varied between 0 and 3 (*Figure 2—figure supplement 3A*). The integral of each gamma kernel was set to 0.02, reflecting the connection strength. Since we added spikes to the network, which increases the average firing rates, we lowered the starting rates by a factor and repeated the process until the average rate resembled the rate before adding the connections. As a criterion for similarity we correlated the randomly drawn rates with the network rates and stopped when the residual error was below 0.005. For the results in *Figure 2—figure supplement 3* and *Figure 4—figure supplement 1* we did not vary the connection strength in order to avoid interaction effects between connection strength and firing rate. However, we varied connection strength randomly between 0.005 and 0.035 with no detectible change to the results. Alternatively, we used a Boxcar kernel (20 ms, integral: 0.02) instead of gamma functions as transfer kernel, which did not degrade the results of this model.

For both sets of networks (SN and CN), ten artificial networks with 100 neurons were calculated and processed identically to the real data. Signal detection theory was used to evaluate detectability of connections based on significant CCH peaks or troughs with the originally modeled networks as a reference. Each pairing was classified into one of four categories: '*Hit*', if a connection was correctly detected, '*Miss*', if a connection was not detected, '*Correct rejection*' (CR), if a non-existing connection was detected as no connection, and '*False Alarm*' (FA), if a non-existing connection was detected as a connection.

## Subsampling model

We generated an artificial neuronal plane with random (Poisson distributed), distance-dependent connectivity density based on our empirically collected data (*Figure 3B*). We modeled 2 cortical areas, each divided into 5 sub-regions coverable by an array, each sub-region covered with 160 electrode positions, and 20 single units per electrode, giving a total of 32,000 neurons. *Figure 4—figure supplement 2A* shows the degree centrality distribution of the full network with an average degree of 3000 and a standard deviation of 70.

Next, we randomly selected 12 subsamples from the neuronal plane with exactly the number of neurons detected as in the real datasets. Subsampling was done with the restriction that always both areas were chosen, with 2 array sub-regions per area and 32 electrode positions per sub-region, reflecting the real recording configuration in most of the datasets. Subsampled networks were then analyzed with the same complex network measures as the real data.

To address the problem that subsampling could artificially cause a heavy tailed degree centrality distribution, even if the underlying connectivity is random, as described in *Han et al. (2005)*, we had a closer look at the parameters mentioned in this study. The average degree of their analyzed networks was 2.19 (SD = 0.45, min = 1.84, max = 2.98), in contrast to our average (non-normalized) degree of 8.28 (SD = 5.73, min = 3.87, max = 25.59). Note that the highest average degree of their analyzed networks was smaller than the lowest average degree of our analyzed networks. More importantly, the underlying networks of their study were strongly fragmented into components (min = 70, max = 591 components), while we excluded all single units which were not part of the largest component, resulting in one component for analysis, while their largest average component size was 20.2. Our network analysis was done on average on 70 single units (min 30, max 148 single units). Based on these different network parameters we concluded that the detected topology, in particular falsely detected power law degree distribution, could be due to the fragmentation into different components. To evaluate this, we created neuronal planes with distance dependent connection density of 1/5, 1/4, 1/3, 1/2, 1, 2, 3, 4, and 5 times of the empirically collected data. After subsampling, we estimated the goodness of fit for the power law model to the degree centrality distribution, the size of the largest component relative to the whole network, and the level of compartmentalization, described by

$$Compartmentalization = \frac{P-1}{N-1} \qquad (12)$$

where $N$ is the number of neurons in the network and P the number of separate components (*Figure 4—figure supplement 2C*).

## Frequency analyses

We estimated the oscillatory behavior of significant connections of single units (according to CCHs) and the spiking of single units themselves (*Bair et al., 1994*; *Mureşan et al., 2008*) (according to ACHs). Since different oscillation frequencies could be present, we computed power spectra of all corrected CCHs and ACHs (*Mureşan et al., 2008*). The power spectrum gives the magnitude of a signal as a function of frequency. To avoid distortions by sharp peaks with small delays that are occasionally present in CCHs (*Fujisawa et al., 2008*), which cause a broad band increase in power due to their impulse like properties, we cut out the time range from $-5$ ms to 5 ms and interpolated the segment linearly. Importantly, sharp peaks were only removed for spectral analyses and not for functional connectivity analyses. Frequency spectra were computed using a discrete Fourier transform algorithm (*Siegel et al., 2009*) (100 logarithmically scaled frequencies from 3 to 100 Hz). Note that computing power spectra of CCHs and ACHs instead of raw spike trains reduced the influence of firing rate on the power spectrum as well as the problem of frequency leakage due to the binary properties of the spike train (*Bair et al., 1994*). In analyzing such a large range of frequencies we had to take the specific characteristics of CCHs into account. Underlying oscillation frequencies in physiology are not phase stable, which leads to a limited number of side lobes in the CCH or ACH. The number of side lobes are also strongly frequency dependent, which makes the ideal window length for Fourier transformation around the 0 time lag frequency dependent. We used Hanning windows of four times the frequency of interest period (with a maximum of 1000 ms and a minimum of 150 ms) aligned on the 0 time bin of the CCHs (*Figure 5—figure supplement 1A*), resulting in approximately 1/frequency and half octave spectro-temporal bandwidth. Each frequency bin was divided by its window length for correct scaling of all frequency bins. To determine significance, we repeated spectral analysis on the corrected surrogate CCHs and ACHs, subtracted their mean spectra from the corresponding spectra of real data and used a cluster-based surrogate test as described before to evaluate the significance of the underlying frequencies in the CCHs.

Spectral analysis of the ACHs differed in one point. Hanning windows covering only one half of the ACHs (with a maximum of 500 ms and a minimum of 75 ms) aligned on the 0 time lag were used (*Figure 5—figure supplement 1B*). By doing so, an accurate measure of the full frequency range with little distortion of refractory effects present in ACHs (*Mureşan et al., 2008*) was obtained.

## Oscillatory vs non-oscillatory synchronization model

We generated pairs of neurons with 600 trials and a trial length of 3.1 s, similar to our recorded data. Spike trains of neurons were generated as a probabilistic process. In case of oscillatory firing neurons, the probability function was a 20 Hz sinusoid. For non-oscillating neurons, we first randomized the 20 Hz sinusoid, in a second step filtered it with a non-causal 50 Hz low-pass filter (Butterworth filter, fourth order) in order to produce a similar decay in spiking probability, and in a last step the filtered probability vector was variance matched with the 20 Hz sinusoid to have a maximum degree matching between the two kinds of probability functions. For each trial the same probability function was used for both neurons with a spiking probability of 0.05 per ms to stay in a physiological range. Independent Poisson distributed noise was added to both neurons representing background stochastic firing, resulting in an average rate of around 5 Hz per neuron. Varying the different parameters within physiological ranges did not alter the results. To simulate different degrees of coupling strengths we systematically varied the trial-wise time offset in spiking of the pair of neurons to each other from completely synchronized to a jitter of a complete cycle (50 ms) in steps of 1 ms.

## Acknowledgements

We thank R Lbik, N Bobb, and S Borchert for animal support, M Dörge for technical support, and S Suway and T Wunderle for help with the development of data analysis and fruitful discussions. This

work was supported by the Deutsche Forschungs Gemeinschaft (SCHE 1575/1-1, SCHE 1575/3-1, and SFB 889, C9), the Bundesministerium für Bildung und Forschung (BCCN-II, 01GQ1005C) and the European Union (FP7-611687, NEBIAS).

## Additional information

### Funding

| Funder | Grant reference number | Author |
|---|---|---|
| Deutsche Forschungsge-meinschaft | SCHE 1575/1-1 | Hansjörg Scherberger |
| European Commission | FP7-611687, NEBIAS | Hansjörg Scherberger |
| Bundesministerium für Bildung und Forschung | BCCN-II, 01GQ1005C | Hansjörg Scherberger |
| Deutsche Forschungsge-meinschaft | Research Group FOR 1847, project B3 (SCHE 1575/3-1) | Hansjörg Scherberger |
| Deutsche Forschungsge-meinschaft | SFB 889, project C9 | Hansjörg Scherberger |

The funders had no role in study design, data collection and interpretation, or the decision to submit the work for publication.

### Author contributions

BD, Conception and design, Acquisition of data, Analysis and interpretation of data, Drafting or revising the article; JAM, Analysis and interpretation of data, Drafting or revising the article; SS, Acquisition of data, Drafting or revising the article; HS, Performed array implantation surgery, Conception and design, Drafting or revising the article

### Author ORCIDs

Benjamin Dann, http://orcid.org/0000-0003-4332-0285
Jonathan A Michaels, http://orcid.org/0000-0002-5179-3181
Stefan Schaffelhofer, http://orcid.org/0000-0002-1006-971X
Hansjörg Scherberger, http://orcid.org/0000-0001-6593-2800

### Ethics

Animal experimentation: All procedures and animal care were conducted in accordance with the guidelines for the care and use of mammals in neuroscience and behavioral research (National Research Council, 2003), and were in agreement with German and European laws governing animal care. Authorization for conducting this study has been granted by the regional government office, the Animal Welfare Division of the Office for Consumer Protection and Food Safety of the State of Lower Saxony, Germany (permit no. 032/09). Monkey handling also followed the recommendations of the Weatherall Report of good animal practice. Animals were pairhoused in a spacious cage (well exceeding legal requirements) and were maintained on a 12-hour on/off lighting schedule. Housing procedures included an environmental enrichment program with access to toys, swings, and hidden treats (e.g., seeds in sawdust). Monkeys had visual and auditory contact to other monkeys. They were fed on a diet of enriched biscuits and fruits. Daily access to fluids was controlled during training and experimental periods to promote behavioral motivation. All surgical procedures were performed under anesthesia, and all efforts were made to minimize post-surgical pain or suffering. Institutional veterinarians continually monitored animal health and well-being.

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
