## [Decision Letter]

Thank you for submitting your article "Uniting functional network topology and oscillations in the fronto-parietal single unit network of behaving primates" for consideration by *eLife*. Your article has been reviewed by three peer reviewers, one of whom, Klaas Stephan (Reviewer #1), is a member of our Board of Reviewing Editors, and the evaluation has been overseen by David Van Essen as the Senior Editor. One other reviewer has agreed to reveal his identity: Nicholas Hatsopoulos (Reviewer #3).

The reviewers have discussed the reviews with one another and the Reviewing Editor has drafted this decision to help you prepare a revised submission.

Summary of manuscript:

This paper presents the results of graph theoretical analyses of a network of single neurons distributed over three cortical areas. Neuronal activity was recorded in three Macaque monkeys while they performed a grasping task. The recording activity was used to infer directed connections amongst pairs of neurons, and the resulting connectivity matrix served as the basis for subsequent analyses. The authors find that this single neuron network is modular and has not only a small world, but also a rich-club structure. Remarkably, rich-club neurons showed oscillatory synchronisation in the beta and low frequency (< 7 Hz) range, while the remaining neurons tended to show non-oscillatory synchrony. These results imply that topological and functional properties of single neurons are related, and that oscillatory and non-oscillatory interactions assume distinct roles at the network level.

Summary of reviews:

All three reviewers were positively impressed by the paper and agreed that it contributes some very interesting findings with potentially wide-ranging implications. However, they also had a number of methodological concerns which would need to be addressed in a revision of your paper. These issues may require additional analyses of your existing data.

The policy of the journal is to provide you with a single set of comments which reflect the consensus view amongst reviewers. These comments can be found below and are divided into essential or Major Issues, which must be addressed convincingly, and Minor Issues. We hope that you will find these comments helpful to further improve the paper.

Major issues

1) The graph theoretical analysis rests on the validity of the underlying connectivity matrix. The latter consists of estimates of directed functional connectivity, obtained from analysing cross-correlation histograms (CCHs) with regard to signatures of temporal precedence, including a correction for common driving input by subtraction of surrogacy CCHs. While CCHs are not an uncommon statistic in electrophysiology, we wondered whether your specific approach to extract directed functional connectivity estimates from CCHs has been examined and validated in previous literature? If so, it would be helpful if the respective references were provided and the evidence for the validity of this approach was summarised in the paper. If not, validation analyses would be required before the graph-theoretical results can be trusted. Ideally, this would involve "ground truth" simulations in which the method is (i) challenged to recover directed functional interactions that are known (face validity) and/or (ii) compared against alternative established measures of directed functional connectivity (construct validity), such as Granger causality.

This issue is important because, as straightforward as the present method may seem, the estimation of directed connectivity is a notoriously difficult issue, and it is now widely accepted that methods of functional/effective connectivity cannot be motivated by theory along but require careful validation.

2) Following on from the previous point, your method removes effects of common stimulus- or movement-locked inputs (although it needs to be clarified what event is used for aligning in order to generate the PSTHs) because the PSTHs reflect these stimulus- or movement-locked inputs. However, the method does not seem to account for common driving input to the two neurons that is not locked to the stimulus or movement onset.

3) Given that CCHs are sensitive to oscillatory behaviour in the neurons studied, the question arises whether this could induce a bias when comparing the connectional properties (derived from the CCHs) between neuronal units that show oscillatory versus non-oscillatory synchrony? Put differently, if estimates of directed functional connectivity were affected by whether or not the neurons in question show oscillatory activity, would this not lead, by construction, to systematic differences in the connectivity patterns of neurons with oscillatory versus non-oscillatory synchrony?

4) One unresolved question is whether the oscillatory correlations are cause or consequence of the rich-club network role of the respective neurons. You might be able to address this to some degree. Several previous studies have emphasized that oscillations in frontal and motor cortices have a transient nature (e.g. Murthy & Fetz J Neurophysiol. 1996; Lundqvist, Miller, Neuron, 2016). You could analyze your LFP recordings to dissociate oscillatory from less oscillatory periods and then investigate whether the same neurons still show the same or different rich-club role.

5) In the ACHs and CCHs, the energy of a given frequency bin is distributed over all time bins, e.g. a beta rhythm leads to multiple peaks and troughs across the width of the ACH or CCH. This diminishes the sensitivity of the statistical testing for oscillatory correlations. It seems possible to improve the analysis by first performing a Fourier transform on the ACHs or CCHs and then performing statistical testing. Currently, you first statistically test ACHs and CCHs in the time domain, and only forward the correlograms with significant clusters to Fourier transformation and statistical testing in the frequency domain. This approach might miss many significant oscillatory correlations.

6) Materials and methods section: Network analysis subsection: Oscillatory synchronization with time lag can result in ambiguous CCHs, in which it is not clear which unit is leading and which is lagging. Can you essentially exclude this ambiguity in your data, e.g. because leads/lags are only a small fraction of the cycle, or because maximal peaks exceeded the next higher peaks substantially?

7) Similarly, it is difficult to understand the interpretation of a cross-correlation peak at time zero as indicative of a bidirectional connection. It seems more likely that this is due to common input from another third neuron. How can one interpret a zero time-lag peak as indicative of bidirectional interactions?

8) You analyze distance dependency by defining distance in categories like “same electrode, “same array" and “same area". In an additional analysis, distance should be defined as actual physical distance. Figure 3 shows that some electrode pairs between F5 and M1 are closer than some other pairs within M1. Physical distance has been shown before to explain a substantial fraction of the variance in neuronal correlation.

9) It would useful to know what kind of time leads/lags were observed among significant cross-correlation histograms. Given that you are looking for significant cross-correlation peaks at time leads/lags spanning +/- 200 ms, if the peaks occurred at very large time leads/lags approaching 200 ms, in what sense are the two neurons interacting physiologically?

10) Material and methods section: Frequency analyses subsection: You remove sharp peaks with small delays with the argument that they would cause distortions. However, such sharp peaks constitute important data and must not be removed, unless they are due to artifacts. If you wish to argue that the sharp peaks are artifacts, evidence needs to be provided. Otherwise, they should be included in the analysis.

---

## [Author Response]

*Major issues*

*1) The graph theoretical analysis rests on the validity of the underlying connectivity matrix. The latter consists of estimates of directed functional connectivity, obtained from analysing cross-correlation histograms (CCHs) with regard to signatures of temporal precedence, including a correction for common driving input by subtraction of surrogacy CCHs. While CCHs are not an uncommon statistic in electrophysiology, we wondered whether your specific approach to extract directed functional connectivity estimates from CCHs has been examined and validated in previous literature? If so, it would be helpful if the respective references were provided and the evidence for the validity of this approach was summarised in the paper. If not, validation analyses would be required before the graph-theoretical results can be trusted. Ideally, this would involve "ground truth" simulations in which the method is (i) challenged to recover directed functional interactions that are known (face validity) and/or (ii) compared against alternative established measures of directed functional connectivity (construct validity), such as Granger causality.*

*This issue is important because, as straightforward as the present method may seem, the estimation of directed connectivity is a notoriously difficult issue, and it is now widely accepted that methods of functional/effective connectivity cannot be motivated by theory along but require careful validation.*

We agree with the reviewers that functional connectivity measures require careful validation, especially directed functional connectivity measures. In many respects, our method for detection of directed functional connectivity is not new, but rather a combination of well described findings using CCHs as a measure. Our method was originally motivated by the work of Moore et al. (1970), who first classified directional influences based on CCHs. However, in that original work peaks with a zero time lag were classified as common drive. In contrast, several modern studies have shown that zero time lag are much more likely to represent true network motives, including reciprocal connections (Vicente et al., 2008; Gollo et al., 2014). Therefore, we classified zero time lag peaks and troughs as bidirectional. To describe our estimation of directed connectivity in more detail, we added a corresponding paragraph in the Results section.

For additional validation of our method we expanded our analyses based on our equal rate model (”Frequency analyses” subsection) to test in detail how well we could (1) detect functional interactions on the basis of realistic cortical firing rates, (2) recover directed functional interactions for different network topologies, and (3) recover these network topologies. We could detect functional connections reasonable well (~64%) and with remarkably few false connections (< 1%) independent of the underlying topology. For the connections detected, we could correctly classify the directionality with an average hit rate of 94% and a correct rejection rate of 74%, with only small differences for different network topologies. It can therefore be concluded that if we detect a connection, our prediction of its directionality is quite accurate. These findings are described in detail in the Results section (”Functional connectivity” subsection) together with an additional supplementary figure (Figure 2—figure supplement 3).

As the most crucial validation of our graph-theoretical results we compared the topology of simulated networks with their detected network topology, hence giving a direct estimate of any possible biases on the network level. We could recover quite well the presence of small-world topology, heavy-tailed, as well as normal degree centrality distribution, the underlying betweenness centrality distribution, and a significant rich-club topology, if present. To illustrate the described topology comparisons, we added an additional supplementary figure (Figure 4—figure supplement 1) and described the results in detail in the Results section (”Degree centrality, betweenness centrality, and hubs” subsection, Results section). This result based on simulations, including realistically low firing rates as found in the cortex, allows us to draw the conclusion that our specific approach to extract directed functional connectivity estimates from CCHs is precise enough to recover the underlying directed functional network topology.

*2) Following on from the previous point, your method removes effects of common stimulus- or movement-locked inputs (although it needs to be clarified what event is used for aligning in order to generate the PSTHs) because the PSTHs reflect these stimulus- or movement-locked inputs. However, the method does not seem to account for common driving input to the two neurons that is not locked to the stimulus or movement onset.*

We agree that common drive not locked to stimulus- or movement-locked inputs is an important topic for functional connectivity measures and should be carefully considered and discussed. We realize that we did not describe in sufficient detail how the surrogate data was created, especially with respect to the fact that the trial-wise firing rates were preserved, which also corrects for trial-wise fluctuations in spiking. We improved the paragraph in the Methods section where the exact procedure is described of how we generated the surrogate data and clarified which events were used as alignments in order to generate PSTHs (Materials and methods section, subsection “Network analysis”).

Nevertheless, there are two possible sources of non-stimulus or movement locked common drive that remain to be discussed. The first is the possibility that two neurons receive input from a third neuron while themselves being functionally uncoupled, resulting in a peak in the CCHs due to their input similarity. We investigated this possibility based on our extended equal rate model, as mentioned in the first point, by explicitly focusing on common drive pairs. For simulated networks with normal distributed connectivity and no other topological features, 0.42% of common drive pairs were detected as significant, while for complex networks with heterogeneous connectivity, small-world and rich-club topology, 1.24% of common drive pairs were detected as significant. We conclude from these results that for low firing regimes and weak pairwise correlations, such as present in the cortical regions from which we recorded, our method for detecting functional connectivity is hardly biased by pairwise common drive, independent of the underlying topology. These results are described in detail in the Results section.

The second possibility is that spatial units such as cortical columns or areas receive non-stimulus or movement locked common drive. However, this situation would result in strong, homogenous connections for close-by units, most likely resulting in a uniformly distributed connectivity pattern across the networks. In contrast, we found strongly heterogeneous networks with neighboring single units showing strong differences in connectivity (Figure 3). Furthermore, the distribution of connections is heavy tailed, which is in conflict with what would be expected by a significant common drive influence. We conclude from these results that the probability for a significant influence of event-unrelated common drive is very low. As an additional point, neurons across the whole network were connected with different latencies, either by beta, low frequency, or non-oscillatory synchrony (Figure 2—figure supplement 4), which can also not be explained by an event-unrelated common drive. To address this point in the text, we added a paragraph to the Discussion section (Subsection “Neural recordings and spike sorting”).

*3) Given that CCHs are sensitive to oscillatory behaviour in the neurons studied, the question arises whether this could induce a bias when comparing the connectional properties (derived from the CCHs) between neuronal units that show oscillatory versus non-oscillatory synchrony? Put differently, if estimates of directed functional connectivity were affected by whether or not the neurons in question show oscillatory activity, would this not lead, by construction, to systematic differences in the connectivity patterns of neurons with oscillatory versus non-oscillatory synchrony?*

The concern that our results could be biased by an increased sensitivity of CCHs to oscillatory behavior was one of our major concerns as well. For this purpose, we first tested for significant peaks or troughs in the CCHs and only in a second step for significant peaks in the Fourier-domain, as described in the methods section. By doing so we made sure that any connection detected was based solely on a significant amount (or suppression) of coincidental spikes without any selective sensitivity for oscillatory couplings.

However, to address this point in a more general fashion we simulated pairs of neurons with either an oscillatory or non-oscillatory firing pattern. For a systematic assessment of possible differences in detecting oscillatory and non-oscillatory synchronization, we varied the offset in spike timing across a physiologically realistic number of simulated trials. Coupling strength measured by CCHs decreased in a similar manner for oscillatory and non-oscillatory firing patterns. We conclude from these results that the sensitivity of the CCH is mainly a function of the constant offset in spike timing between two neurons and does not depend on whether the underlying firing pattern is oscillatory or not. To illustrate the described simulation, we added a supplementary figure (Figure 5—figure supplement 2) and describe the simulation in the Methods (Subsection “Oscillatory vs non-oscillatory synchronization model”) and Results (Subsection “Network topology of oscillatory synchrony”) sections.

*4) One unresolved question is whether the oscillatory correlations are cause or consequence of the rich-club network role of the respective neurons. You might be able to address this to some degree. Several previous studies have emphasized that oscillations in frontal and motor cortices have a transient nature (e.g. Murthy & Fetz J Neurophysiol. 1996; Lundqvist, Miller, Neuron, 2016). You could analyze your LFP recordings to dissociate oscillatory from less oscillatory periods and then investigate whether the same neurons still show the same or different rich-club role.*

We agree that this is a very interesting question worth addressing. Since a minimum number of trials are required to properly estimate the functional connectivity for topological analyses we used the two datasets from monkey M containing more than 900 trials (see Table 1). Instead of using our LFP recordings to dissociate periods of high and low levels of oscillatory synchrony, we calculated single trial population ACH spectra. In order to avoid bias by the time course of the trial or different conditions, we split the data based on the amount of single trial beta and low frequency power into two blocks with an equal number of trials per condition, and repeated our functional connectivity and topological analyses. Interestingly, even though the unit-wise degree and the degree distribution was hardly different between the two networks, the high oscillatory state network showed a clear rich-club topology, while the low oscillatory state network hardly showed any rich-club effect, similar for both datasets. These results suggest that the oscillatory synchrony and functional rich-club topology we observed are directly linked. The described findings are presented in an additional supplementary figure (Figure 5—figure supplement 3) and described in detail in the Results section (Subsection “Network topology of oscillatory synchrony”).

*5) In the ACHs and CCHs, the energy of a given frequency bin is distributed over all time bins, e.g. a beta rhythm leads to multiple peaks and troughs across the width of the ACH or CCH. This diminishes the sensitivity of the statistical testing for oscillatory correlations. It seems possible to improve the analysis by first performing a Fourier transform on the ACHs or CCHs and then performing statistical testing. Currently, you first statistically test ACHs and CCHs in the time domain, and only forward the correlograms with significant clusters to Fourier transformation and statistical testing in the frequency domain. This approach might miss many significant oscillatory correlations.*

We agree that it is possible that we missed some oscillatory connections where the energy is widely distributed over all time bins, but this could be also the case for any irregular non-oscillatory temporal pattern, such as broad peaks or troughs not exceeding the selection criterion (for an example see: Figure 2 bottom panel). As we illustrated in Figure 5—figure supplement 2, the sensitivity of the CCH is mainly a function of the constant offset in spike timing between two neurons and not whether the synchronization is oscillatory or not (see response to issue 3).

All ACHs were tested in the frequency domain, since the time domain is irrelevant for connectivity and we were only interested in their oscillatory properties to classify single units into oscillators and non-oscillators. Therefore, we can be sure that no oscillatory patterns were missed for ACHs. To clarify this point, we changed a sentence in the Results section (Subsection “Network topology of oscillatory synchrony”).

The main reason why we first tested CCHs in the time domain, and only forward those CCHs with significant clusters to the Fourier transform and statistical testing in the frequency domain, was to be sure to not bias sensitivity towards oscillatory behavior (see issue 3).

*6) Materials and methods section: Network analysis subsection: Oscillatory synchronization with time lag can result in ambiguous CCHs, in which it is not clear which unit is leading and which is lagging. Can you essentially exclude this ambiguity in your data, e.g. because leads/lags are only a small fraction of the cycle, or because maximal peaks exceeded the next higher peaks substantially?*

Thank you for pointing out this possible ambiguity. To address this point we added a histogram showing the relative peak phase to the zero time lag separate for both frequencies present in the data (beta: 20Hz, and low frequency oscillations: 4Hz; Figure 2—figure supplement 4), and we describe these findings in an added paragraph in the Results section (Subsection “Functional connectivity”). For most of the CCHs showing oscillatory synchrony, the maximum peak phase lags were within half a cycle of the corresponding frequency (beta connections: 77.70% < π, low frequency connections: 87.66% < π), with a maximum around zero phase lag synchrony for both frequencies. These results suggest that we can properly evaluate the directional influence of oscillatory synchronized connections in our networks. Furthermore, it is important to state, that all our results for degree centrality, including the rich-club coefficient, were based on the total degree (sum of in-degree and out-degree), and were therefore unaffected by the directionality of oscillatory synchrony.

*7) Similarly, it is difficult to understand the interpretation of a cross-correlation peak at time zero as indicative of a bidirectional connection. It seems more likely that this is due to common input from another third neuron. How can one interpret a zero time-lag peak as indicative of bidirectional interactions?*

The classification of zero time lag synchronization is an important topic and classically thought to represent common drive (Moore et al., 1970). However, several modern studies showed that zero time lag synchronization can be mainly explained by reciprocal/bidirectional connections (Vicente et al., 2008; Gollo et al., 2014). The underlying assumed mechanism is dynamical relaying. To better address this point we added a paragraph to the Results section (Subsection “Functional connectivity”), in accordance with the response to issue 1.

Furthermore, we explicitly checked the properties of common drive pairs and bidirectional connections based on our equal rate model simulation datasets (Figure 2—figure supplement 3) as described in detail in the Results section (Subsection “Functional connectivity”) and in accordance with our response to issue 2. The average CCH peak amplitude of bidirectional pairs was on average 24 times higher than that of common drive pairs, with a maximum at the zero time lag. When we explicitly checked the distribution of maximum peak lags we found that bidirectional connections had their maximum peak at the zero time lag 7.38 times more often than common drive pairs (for a window of +/- 2 ms around the zero time lag 9 times more often). In summary, our simulations revealed a much higher likelihood that zero time lag peaks represent bidirectional connections, rather than common drive, in line with the simulations described in the papers mentioned.

*8) You analyze distance dependency by defining distance in categories like “same electrode, “same array" and “same area". In an additional analysis, distance should be defined as actual physical distance. Figure 3 shows that some electrode pairs between F5 and M1 are closer than some other pairs within M1. Physical distance has been shown before to explain a substantial fraction of the variance in neuronal correlation.*

Thank you for your accurate observation, since in monkey M we had electrodes implanted in area M1, some electrodes in F5 and M1 are closer to each other than to electrodes within the same area. For all three datasets recorded from monkey M we estimated physical distance based on the anatomical diagram (Figure 1) and defined distance groups and repeated all statistics for network measures, as suggested. All statistics for network measures yielded nearly identical results, with no case where a measure was significant when it was not for categorical distance dependent shuffling, and vice versa for non-significant measures, as described in detail in the Methods section (Subsection “Statistics for network measures”).

*9) It would useful to know what kind of time leads/lags were observed among significant cross-correlation histograms. Given that you are looking for significant cross-correlation peaks at time leads/lags spanning +/- 200 ms, if the peaks occurred at very large time leads/lags approaching 200 ms, in what sense are the two neurons interacting physiologically?*

We agree that it is important to identify the precise lead/lag times for all CCHs in order to classify their physiological meaning. For this purpose, we added a histogram showing maximum peak time lags relative to the zero time lag for all CCHs (Figure 2—figure supplement 4). Interestingly, most troughs/peaks had very short time lags (45.67% < 10 ms, and 85.12% < 100 ms) in agreement with what is physiologically expected. To describe these results in the text, we added a paragraph in the Results section (Subsection “Functional connectivity”).

*10) Material and methods section: Frequency analyses subsection: You remove sharp peaks with small delays with the argument that they would cause distortions. However, such sharp peaks constitute important data and must not be removed, unless they are due to artifacts. If you wish to argue that the sharp peaks are artifacts, evidence needs to be provided. Otherwise, they should be included in the analysis.*

We indeed removed sharp peaks with small delays with the argument that they would cause distortions, but only during spectral analyses. For the detection of significant connections, we did not remove the sharp peaks. It is important to state that a large proportion of our found connections were based on sharp peaks. Nevertheless, for spectral analyses they cause a broad band increase in power due to their impulse-like properties, which does not reflect any oscillatory synchrony, and were therefore removed sharp peaks for spectral analyses. For clarification, we added a sentence to the Methods section.